# A novel heteromeric pantothenate kinase complex in apicomplexan parasites

Erick T. Tjhin[1ʘ], Vanessa M. Howieson[1ʘ], Christina Spry[1], Giel G. van Dooren[1], Kevin J. Saliba[1,2]*

1 Research School of Biology, The Australian National University, Canberra, Australia, 2 Medical School, The Australian National University, Canberra, Australia

ʘ These authors contributed equally to this work.
* kevin.saliba@anu.edu.au

## Abstract

Coenzyme A is synthesised from pantothenate via five enzyme-mediated steps. The first step is catalysed by pantothenate kinase (PanK). All PanKs characterised to date form homodimers. Many organisms express multiple PanKs. In some cases, these PanKs are not functionally redundant, and some appear to be non-functional. Here, we investigate the PanKs in two pathogenic apicomplexan parasites, *Plasmodium falciparum* and *Toxoplasma gondii*. Each of these organisms express two PanK homologues (PanK1 and PanK2). We demonstrate that *Pf*PanK1 and *Pf*PanK2 associate, forming a single, functional PanK complex that includes the multi-functional protein, *Pf*14-3-3I. Similarly, we demonstrate that *Tg*PanK1 and *Tg*PanK2 form a single complex that possesses PanK activity. Both *Tg*PanK1 and *Tg*PanK2 are essential for *T. gondii* proliferation, specifically due to their PanK activity. Our study constitutes the first examples of heteromeric PanK complexes in nature and provides an explanation for the presence of multiple PanKs within certain organisms.

## Author summary

Apicomplexans are a phylum of obligate intracellular parasites that cause diseases in humans and other animals, inflicting considerable burdens on human societies. During their intracellular stage, these parasites must scavenge vitamins from their host organisms in order to survive and proliferate. One such vitamin is pantothenate (vitamin $B_5$), which parasites convert in a universal five-step pathway to the essential cofactor coenzyme A (CoA). The first reaction in the CoA biosynthesis pathway is catalysed by the enzyme pantothenate kinase (PanK). The genomes of humans and many other organisms, including apicomplexans, encode multiple PanK homologues, although in all studied examples, the functional PanK enzyme exists as a homodimer. In this study, we demonstrate that the two PanK homologues encoded in the genomes of the apicomplexans *Plasmodium falciparum* and *Toxoplasma gondii*, PanK1 and PanK2, exist as functional heteromeric complexes. We provide evidence that both PanK homologues contribute to the PanK activity in these parasites, and that both PanK1 and PanK2 are essential for the proliferation of *T. gondii* parasites specifically for their PanK activity. Our data describe the first known

**Data Availability Statement:** All relevant data are within the manuscript and its Supporting Information files.

**Funding:** ETT, VMH and CS were supported by Research Training Program scholarships from the

Australian Government. CS was also funded by an NHMRC Overseas Biomedical Fellowship (1016357). This work was, in part, supported by a Project Grant (APP1129843) from the National Health and Medical Research Council to KJS and a Discovery Grant (DP150102883) from the Australian Research Council to GvD. The funders had no role in study design, data collection and analysis, decision to publish, or preparation of the manuscript.

**Competing interests:** The authors have declared that no competing interests exist.

instances of heteromeric PanK complexes in nature and may explain why some organisms that express multiple PanKs harbour seemingly non-functional isoforms.

## Introduction

Coenzyme A (CoA) is an essential enzyme cofactor in all living organisms [1]. CoA itself, and the CoA-derived phosphopantetheine prosthetic group required by various carrier proteins, function as acyl group carriers and activators in key cellular processes such as fatty acid biosynthesis, β-oxidation and the citric acid cycle. Pantothenate kinase (PanK) catalyses the first step in the conversion of pantothenate (vitamin $B_5$) to CoA [2]. PanKs are categorised into three distinct types—type I, II and III—based on their primary sequences, structural fold, enzyme kinetics and inhibitor sensitivity. PanKs from all three types have been shown to exist as homodimers based on their solved protein structures [3–10]. All eukaryotic PanKs that have been characterised so far are type II PanKs. Interestingly, many eukaryotes (such as *Arabidopsis thaliana* [11,12], *Mus musculus* [13–16] and *Homo sapiens* [17–21]) express multiple PanKs and in some cases it is clear that these PanKs are not functionally redundant [15,22]. For example, mutations in only one of four type II PanKs in humans causes a neurodegenerative disorder known as PanK-associated neurodegeneration [17]. Some bacteria also express multiple PanKs. For example, some *Mycobacterium* [23], *Streptomyces* [7] and *Bacillus* [7,24,25] species have both type I and type III PanKs, while a select few bacilli (including the category A biodefense pathogen *Bacillus anthracis*) carry both a type II and type III PanK [7]. In some organisms harbouring multiple PanKs, it has not been possible to demonstrate functional activity for all enzymes. One of the four human type II PanKs was shown to be catalytically inactive [21,26], as is a type III PanK from *Mycobacterium tuberculosis* [23], and a type II PanK from *B. anthracis* [7]. The reason for the presence of multiple PanKs within certain cells, and the apparent inactivity of certain PanKs, is unclear.

Two putative genes coding for PanK enzymes have been identified in each of the genomes of the pathogenic apicomplexan parasites *Plasmodium falciparum* (PF3D7_1420600 (*Pfpank1*) and PF3D7_1437400 (*Pfpank2*)) and *Toxoplasma gondii* (TGME49_307770 (*Tgpank1*) and TGME49_235478 (*Tgpank2*)). We have recently shown that mutations in *Pf*PanK1 alter PanK activity in *P. falciparum*, providing evidence that *Pf*PanK1 is an active PanK, at least in the disease-causing stage of the parasite's lifecycle [27]. The function of *Pf*PanK2 and its contribution to PanK activity in *P. falciparum* is unknown. *Pf*PanK2 contains a unique, large insert in a loop associated with the dimerisation of PanKs in their native conformation [8] and this may affect its ability to form a dimer, rendering it inactive [28]. No functional information is available on the putative *T. gondii* PanKs, but a genome-wide CRISPR-Cas9 screen of the *T. gondii* genome predicted that *both* PanK genes are important for parasite proliferation *in vitro* [29]. Similarly, a recent genome-wide insertional mutagenesis study of *P. falciparum* has predicted that mutations in *either* *Pf*PanK1 or *Pf*PanK2 result in significant fitness costs to the parasite [30]. These results suggest that the PanK2 proteins of these parasites play important role(s), although their exact function remains unclear.

In this study, we demonstrate that PanK1 and PanK2 from both *P. falciparum* and *T. gondii* are part of the same, multimeric protein complex in these parasites. This constitutes the first identification of heteromeric PanK complexes in nature. Furthermore, our data provide the first evidence that PanK2 contributes to, and is essential for, PanK function in apicomplexans.

## Results

### *Pf*PanK1 and *Pf*PanK2 are part of the same protein complex

The importance and role of PanK2 in apicomplexan parasites have not previously been established. To characterise the *P. falciparum* PanK2 homologue (*Pf*PanK2), we first determined where in the parasite the protein localises. We episomally expressed *Pf*PanK2-GFP in asexual blood stage *P. falciparum* parasites and found that *Pf*PanK2-GFP is localised throughout the parasite cytosol and is not excluded from the nucleus (**Fig 1A**). This is a similar localisation to

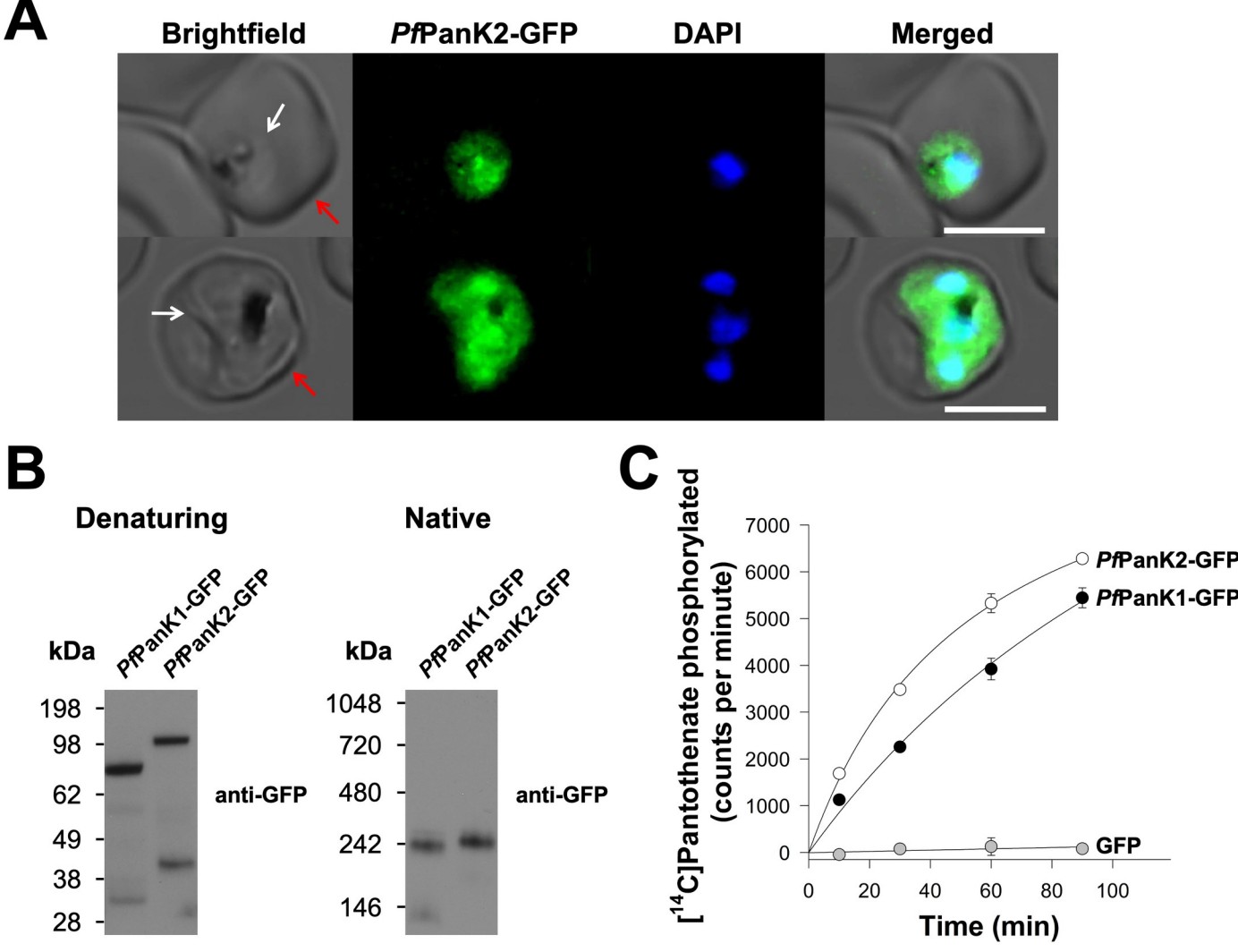

**Fig 1. *Pf*PanK1 and *Pf*PanK2 are part of similar-sized protein complexes that possess PanK activity.** (A) Confocal micrographs showing the subcellular location of *Pf*PanK2-GFP within trophozoite/schizont-stage *P. falciparum*-infected erythrocytes. The nuclei of the parasites are stained with DAPI. From left to right: Brightfield, GFP-fluorescence, DAPI-fluorescence, and merged images. Arrows indicate the plasma membranes of the erythrocyte (red) or the parasite (white). Scale bars represent 5 μm. (B) Denaturing and native western blot analyses of the GFP-tagged proteins in the *Pf*PanK1-GFP and *Pf*PanK2-GFP parasite lines. The expected sizes of the proteins are ~87 kDa for *Pf*PanK1-GFP and ~118 kDa for *Pf*PanK2-GFP. For reference, the molecular mass of the GFP tag is ~27 kDa. Western blots were performed with anti-GFP antibodies and each of the blots shown is representative of three independent experiments, each performed with a different batch of parasites. (C) The phosphorylation of [$^{14}$C]pantothenate (initial concentration of 2 μM, ~10,000 counts per minute) over time by the immunopurified complex from lysates of parasites expressing *Pf*PanK1-GFP (black circles), *Pf*PanK2-GFP (white circles) and untagged GFP (grey circles). Data shown are representative of two independent experiments, each performed with a different batch of parasites and carried out in duplicate. Error bars represent range/2 and are not shown if smaller than the symbols.

what we observed for *Pf*PanK1-GFP previously [27]. Western blotting (**S1 Fig**, **red arrow**) of proteins separated by SDS-PAGE revealed that *Pf*PanK2-GFP has a molecular mass consistent with the predicted mass of the fusion protein (~118 kDa; **Fig 1B**), which is slightly larger than the predicted mass of *Pf*PanK1-GFP (~87 kDa; **Fig 1B**). As PanKs from other organisms exist as homodimers, we undertook blue native-PAGE to determine whether *Pf*PanK1-GFP and *Pf*PanK2-GFP exist in protein complexes. Interestingly, under native conditions, both *Pf*PanK1-GFP and *Pf*PanK2-GFP were found to be part of complexes that are ~240 kDa in mass (**Fig 1B**).

To determine the activity and protein composition of these complexes, we set out to purify the *Pf*PanK1-GFP and *Pf*PanK2-GFP complexes by immunoprecipitation. As a control, we also purified untagged GFP. We verified that most of the GFP-tagged proteins were captured from the total lysates prepared from cell lines expressing the different proteins, with bands corresponding to *Pf*PanK1-GFP, *Pf*PanK2-GFP and the untagged GFP epitope tag detected in the bound fraction of the respective cell lines (**S2 Fig**). To determine whether the purified *Pf*PanK1 and *Pf*PanK2 complexes possess PanK activity, we performed a [$^{14}$C]pantothenate phosphorylation assay (**S1 Fig**, **orange arrow**). We found that 50 – 60% of the [$^{14}$C]pantothenate initially present in the reaction was phosphorylated within 90 min by the immunopurified complex from *both* the *Pf*PanK1-GFP and *Pf*PanK2-GFP lines (**Fig 1C**). Conversely, the immunopurified untagged GFP did not display PanK activity (**Fig 1C**). These experiments provide the first indication that *Pf*PanK1 and *Pf*PanK2 exist as part of an active PanK enzyme complex of similar mass in *P. falciparum* parasites. They also provide the first indication that *Pf*PanK2 contributes to PanK activity in these parasites.

To elucidate the protein composition of the *Pf*PanK1-GFP and *Pf*PanK2-GFP complexes, the immunoprecipitated samples were subjected to mass spectrometry (MS)-based proteomic analysis (**S1 Fig**, **green arrow**; bound fractions of untagged GFP-expressing and 3D7 wild-type parasites were included as negative controls). Both *Pf*PanK1 (36–50% coverage, **S3 Fig**) and *Pf*PanK2 (29–49% coverage, **S4 Fig**) were unequivocally detected as the two most abundant proteins in the immunopurified complex from *both* the *Pf*PanK1-GFP and *Pf*PanK2-GFP lines (**Fig 2A**). Interestingly the next most abundant protein detected in both complexes was *Pf*14-3-3I (43–67% coverage, **Figs 2A** and **S5**). These results are consistent with *Pf*PanK1, *Pf*PanK2 and *Pf*14-3-3I being part of the same protein complex. Other proteins, such as M17 leucyl aminopeptidase (fourth most abundant), were also detected in the MS analysis, albeit with a comparatively fewer number of peptides (**Fig 2A** and **S1 Table**).

Our group had previously generated three mutant strains, termed PanOH-A, PanOH-B and CJ-A, by drug-pressuring *P. falciparum* parasites with antiplasmodial pantothenate analogues [27]. These strains harbour mutations in *Pf*PanK1 (D507N, ΔG95 and G95A, respectively) that affect *Pf*PanK catalytic activity [27]. To test further whether *Pf*PanK1 and *Pf*PanK2 are part of the same protein complex, we introduced episomally-expressed *Pf*PanK2-GFP into these parasite strains (the newly generated lines are designated with "+*Pf*PanK2-GFP" superscript). We then immunopurified the *Pf*PanK2-GFP complex from the PanOH-A$^{+Pf\text{PanK2-GFP}}$, PanOH-B$^{+Pf\text{PanK2-GFP}}$ and CJ-A$^{+Pf\text{PanK2-GFP}}$ lines, as well as from the wild type control (Parent$^{+Pf\text{PanK2-GFP}}$), and performed [$^{14}$C]pantothenate phosphorylation assays with the complex derived from immunopurified *Pf*PanK2-GFP (**S1 Fig**, **orange arrow**). As we reported previously, the *Pf*PanK1 mutations alter the *Pf*PanK activity of PanOH-A, PanOH-B and CJ-A parasites (using lysate, **S1 Fig**, **blue arrow**) such that the following rank order of enzyme activity relative to the Parent line is observed: PanOH-A > Parent > PanOH-B > CJ-A (**Fig 2B(i)**, [27]). Notably, PanK activity of the complex immunopurified from the various *Pf*PanK2-GFP-expressing mutant lines followed the same rank order: PanOH-A$^{+Pf\text{PanK2-GFP}}$ > Parent$^{+Pf\text{PanK2-GFP}}$ > PanOH-B$^{+Pf\text{PanK2-GFP}}$ > CJ-A$^{+Pf\text{PanK2-GFP}}$ (**Fig 2B(ii)**). This difference in

## A

| | No of peptides detected (> 95% confidence) | | | |
| --- | --- | --- | --- | --- |
| | *Pf*PanK1-GFP line co-immunoprecipitation | | *Pf*PanK2-GFP line co-immunoprecipitation | |
| Protein detected | 1st rep | 2nd rep | 1st rep | 2nd rep |
| *Pf*PanK2 | 10 | 24 | 16 | 77 |
| *Pf*PanK1 | 20 | 23 | 19 | 49 |
| 14-3-3 protein (*Pf*14-3-3I) | 7 | 14 | 16 | 41 |
| M17 leucyl aminopeptidase | 3 | 4 | 5 | 8 |

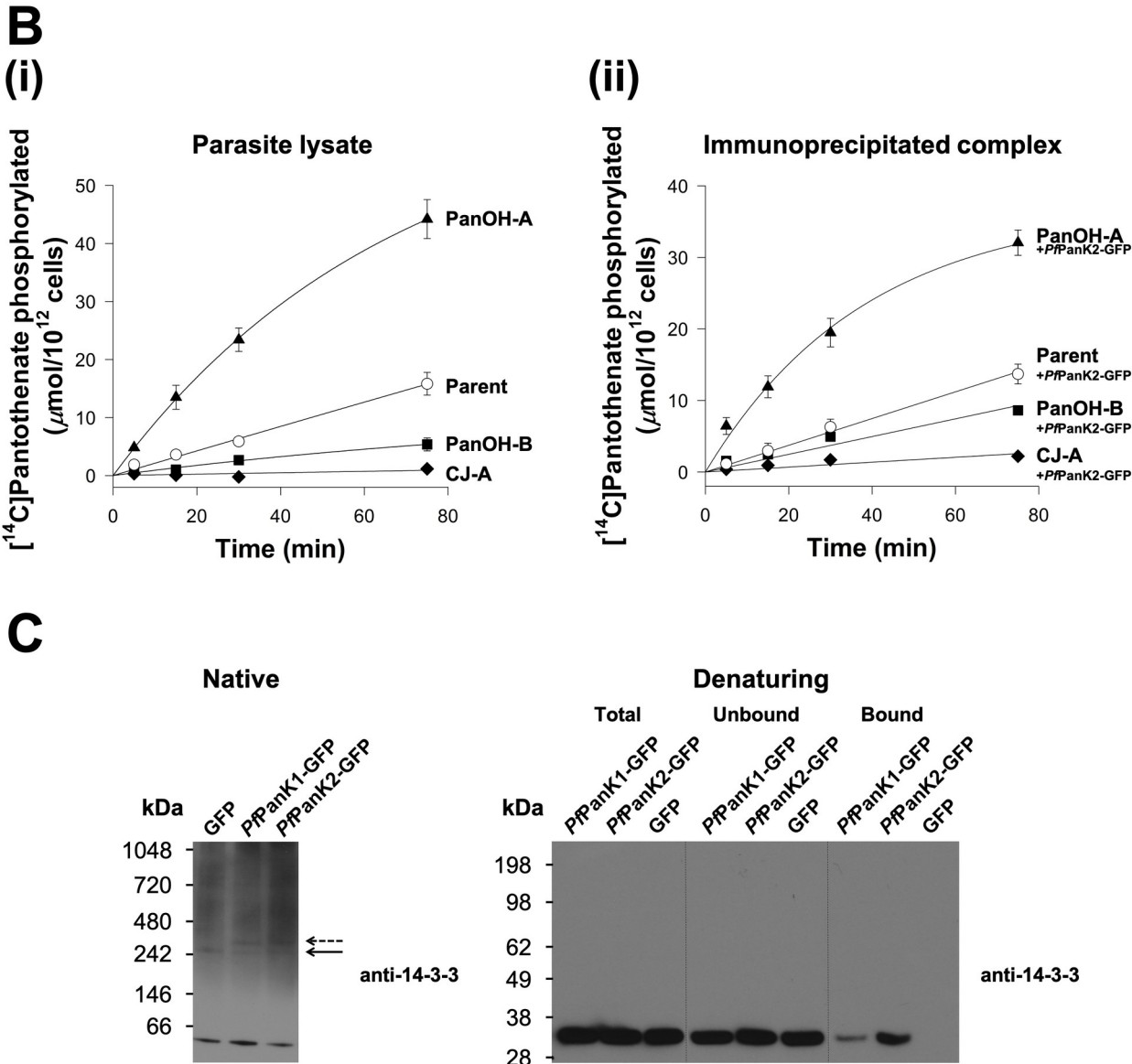

**Fig 2. *Pf*PanK1 and *Pf*PanK2 are part of a single PanK complex that includes *Pf*14-3-3I.** (A) The four most abundant proteins identified in the MS analysis of proteins immunoprecipitated with GFP-Trap from the *Pf*PanK1-GFP and *Pf*PanK2-GFP lines. Data shown are representative of two independent analyses (1st and 2nd rep), each performed with a different batch of parasites. Proteins detected in the untagged GFP line or wild-type 3D7 parasite immunoprecipitations (negative controls) were removed. Only proteins with three or more peptides detected in *both* replicate co-immunoprecipitation experiments are shown. Proteins identified but which did not meet these criteria are shown in S1 Table. Proteins are listed in descending order according to the total number of peptides detected across all replicates (total peptides in all four columns). (B) The phosphorylation of [14C]pantothenate (initial concentration of 2 μM) over time by (i) lysates generated from Parent (white circles), PanOH-A (black triangles), PanOH-B (black squares) and CJ-A (black diamonds) parasite lines (reproduced from [27]) and (ii) proteins immunoprecipitated with GFP-Trap from Parent$^{+Pf\text{PanK2-GFP}}$ (white circles), PanOH-A$^{+Pf\text{PanK2-GFP}}$ (black triangles), PanOH-B$^{+Pf\text{PanK2-GFP}}$ (black squares) and CJ-A$^{+Pf\text{PanK2-GFP}}$ (black diamonds) parasite lysates. Values in (ii) are averaged from three independent experiments, each performed with a different batch of parasites and carried out in duplicate. Error bars represent SEM and are not shown if smaller than the symbols. (C) Native western blot analysis of the lysates and denaturing western blot analyses of the different GFP-Trap immunoprecipitation fractions generated from *Pf*PanK1-GFP and *Pf*PanK2-GFP parasite lines, with the untagged GFP line as a control. Protein samples used in the denaturing western blot were derived from the same immunoprecipitation depicted in S2 Fig. Western blots were performed with pan-specific anti-14-3-3 antibodies (previously shown to detect *Plasmodium* 14-3-3 [32]). Arrows indicate the position of 14-3-3-containing complexes of comparable masses in all three lines (solid arrow) and the complexes found only in the *Pf*PanK1-GFP and *Pf*PanK2-GFP lines (dashed arrow). The native blot shown is a representative of three independent experiments, while the denaturing blot is a representative of two independent experiments, each performed with a different batch of parasites.

pantothenate phosphorylation rates was not due to variations in the amount of *Pf*PanK2-GFP protein in the immunopurified complexes used for the assays (**S6 Fig**). Further, the initial rate of pantothenate phosphorylation by the Parent lysate was 0.211 ± 0.026 (mean ± SEM) μmol/$10^{12}$ cells/min (**Fig 2B(i)**) while that of the immunopurified complex from the Parent$^{+Pf\text{-}}$$^{\text{PanK2-GFP}}$ line was 0.177 ± 0.016 (mean ± SEM) μmol/$10^{12}$ cells/min (**Fig 2B(ii)**), demonstrating that the PanK activity in the immunopurified complexes is indistinguishable (95% CI = -0.119 to 0.051; include 0) from that observed in parasite lysates. Overall, the data shown in **Fig 2B** are consistent with *Pf*PanK2-GFP associating with the mutant *Pf*PanK1 from each cell line and indicate that both proteins are part of the same PanK complex in *P. falciparum*.

Our proteomic analysis identified *Pf*14-3-3I as being co-immunoprecipitated with both *Pf*PanK1 and *Pf*PanK2 (**Fig 2A**). To test whether *Pf*14-3-3I is a *bona fide* component of the PanK complex of *P. falciparum*, we performed western blotting with a pan-specific anti-14-3-3 antibody. Under native conditions (**S1 Fig**, **red arrow**), the 14-3-3 antibody detected a major protein band at <66 kDa (**Fig 2C**), which likely represents dimeric *Pf*14-3-3I of the parasite [31]. We also observed a protein complex of ~240 kDa in the *Pf*PanK1-GFP, *Pf*PanK2-GFP and untagged GFP lines (solid arrow, **Fig 2C**). In addition, a protein complex of slightly higher molecular mass, likely corresponding to the PanK complex that includes the GFP epitope tag, was also observed in the *Pf*PanK1-GFP and *Pf*PanK2-GFP lines but not the untagged GFP line (dashed arrow, **Fig 2C**). As a direct test for whether *Pf*14-3-3I exists in the same complex as *Pf*PanK1 and *Pf*PanK2, we performed western blotting on proteins immunopurified with anti-GFP antibodies from the *Pf*PanK1-GFP, *Pf*PanK2-GFP and untagged GFP parasite lines (**S1 Fig**, **purple arrow**). We found that *Pf*14-3-3I protein was detected in the immunopurified complex from both the *Pf*PanK1-GFP and *Pf*PanK2-GFP lines, but not in that purified from parasites expressing untagged GFP (**Fig 2C**). Together with the native western blot (**Fig 1B**) and proteomic (**Fig 2A**) analyses, these results are consistent with *Pf*PanK1 and *Pf*PanK2 being part of the *same* complex that also contains *Pf*14-3-3I, and that this complex is responsible for the PanK activity observed in the intraerythrocytic stage of *P. falciparum*.

## *Tg*PanK1 and *Tg*PanK2 also constitute a single complex with PanK activity that is essential for parasite proliferation

Based on sequence similarity, *Tg*PanK1 and *Tg*PanK2 are homologous to their *P. falciparum* counterparts (**S7 Fig**). To begin characterising *Tg*PanK1 and *Tg*PanK2, we introduced the coding sequence for a mini-Auxin-Inducible Degron (mAID)-haemagglutinin (HA) tag into the

3' region of the open reading frames of *Tg*PanK1 or *Tg*PanK2 in RH TATiΔ*Ku80*:TIR1 strain *T. gondii* parasites [33] also expressing a 'tdTomato' red fluorescent protein (RFP) (**S8A Fig**). Successful integration of the mAIDHA tag was verified by PCR (**S8B Fig**). Western blotting (**S1 Fig**, **red arrow**) revealed that the *Tg*PanK1-mAIDHA and *Tg*PanK2-mAIDHA proteins have molecular masses of ~160 and ~200 kDa, respectively (**Fig 3A**), corresponding to the predicted sizes of *Tg*PanK1-mAIDHA (143 kDa) and *Tg*PanK2-mAIDHA (189 kDa). When analysed under native conditions, *Tg*PanK1-mAIDHA and *Tg*PanK2-mAIDHA *both* exist in protein complexes of ~720 kDa in mass (**Fig 3A**).

To investigate if *Tg*PanK1 and *Tg*PanK2 are part of the same ~720 kDa complex, we introduced a sequence encoding a GFP tag into the genomic locus of *Tgpank1* in the *Tg*PanK2-mAIDHA strain (**S8A and S8C Fig**). Co-immunoprecipitation experiments (**S1 Fig**, **purple arrow**) revealed that *Tg*PanK1-GFP (160 kDa) co-purified with *Tg*PanK2-mAIDHA (**Fig 3B**). Analogous experiments with a *Tg*PanK1-HA/*Tg*PanK2-GFP line, wherein we integrated a sequence encoding a GFP tag into the *Tgpank2* locus and a sequence encoding a HA tag into the *Tgpank1* locus (**S8A and S8D Fig**), yielded similar results (**S9 Fig**). We therefore conclude that, like *Pf*PanK1 and *Pf*PanK2 in *P. falciparum* (**Figs 1** and **2**), *Tg*PanK1 and *Tg*PanK2 are components of the same protein complex. We also tested whether the PanK complex in *T. gondii* contains an orthologue of 14-3-3I. Western blot analysis (**S1 Fig**, **purple arrow**) of the fractions from *Tg*PanK1-GFP/*Tg*PanK2-mAIDHA co-immunoprecipitation using a pan-14-3-3 antibody showed that *Tg*14-3-3 could be detected in *T. gondii* parasite lysates but not in the *Tg*PanK complex (**S10 Fig**).

To determine whether the *Tg*PanK1/*Tg*PanK2 complex has pantothenate kinase activity, we immunopurified proteins from *Tg*PanK1-GFP/*Tg*PanK2-mAIDHA, *Tg*PanK1-HA/*Tg*PanK2-GFP and control (expressing untagged GFP) cell lines using GFP-Trap, and measured the ability of the purified proteins to phosphorylate pantothenate (**S1 Fig**, **orange arrow**). The samples purified from the *Tg*PanK1-GFP/*Tg*PanK2-mAIDHA and *Tg*PanK1-HA/*Tg*PanK2-GFP lines exhibited higher pantothenate phosphorylation activity than that from the control parasites expressing untagged GFP (**Fig 3C**). These findings indicate that, like the *P. falciparum* PanK complex (**Figs 1** and **2**), the *Tg*PanK1/*Tg*PanK2 complex possesses PanK activity.

Active PanK proteins from other organisms contain conserved nucleotide binding motifs [34,35]. As is the case for *Pf*PanK2, the nucleotide-binding motifs of *Tg*PanK2 deviate substantially from those of other eukaryotic PanKs (**S7 Fig**). It is therefore unclear whether pantothenate phosphorylation is catalysed solely by *Tg*PanK1 or if *Tg*PanK2 also contributes to PanK activity. To answer this, we first investigated whether *Tg*PanK1 and *Tg*PanK2 are important for parasite proliferation (**S1 Fig**, **pink arrows**). *Tg*PanK1 and *Tg*PanK2 were individually knocked down by exposing the mAID-regulated lines to 100 μM indole-3-acetic acid (IAA–a plant hormone of the auxin class), a concentration that we determined was not detrimental to wild-type parasite proliferation. *Tg*PanK1-mAIDHA and *Tg*PanK2-mAIDHA were degraded within an hour of exposing the parasites to IAA (**Fig 4A**). Both the *Tg*PanK1-mAIDHA and *Tg*PanK2-mAIDHA lines express RFP, which enabled us to monitor parasite proliferation using fluorescence growth assays, as described previously [36]. We measured proliferation of the *Tg*PanK1-mAIDHA, *Tg*PanK2-mAIDHA and parental lines cultured in the presence or absence of 100 μM IAA over seven days. In the absence of IAA, we observed a normal sigmoidal growth curve for all three lines (**Fig 4B**). By contrast, we observed a complete cessation of proliferation of the parasite lines expressing *Tg*PanK1-mAIDHA and *Tg*PanK2-mAIDHA, but not the parental strain, in the presence of 100 μM IAA (**Fig 4B**). These data indicate that both *Tg*PanK1 and *Tg*PanK2 are crucial for *T. gondii* proliferation and, notably, that neither can substitute for the other. To establish whether *Tg*PanK1 and *Tg*PanK2 are essential due to the

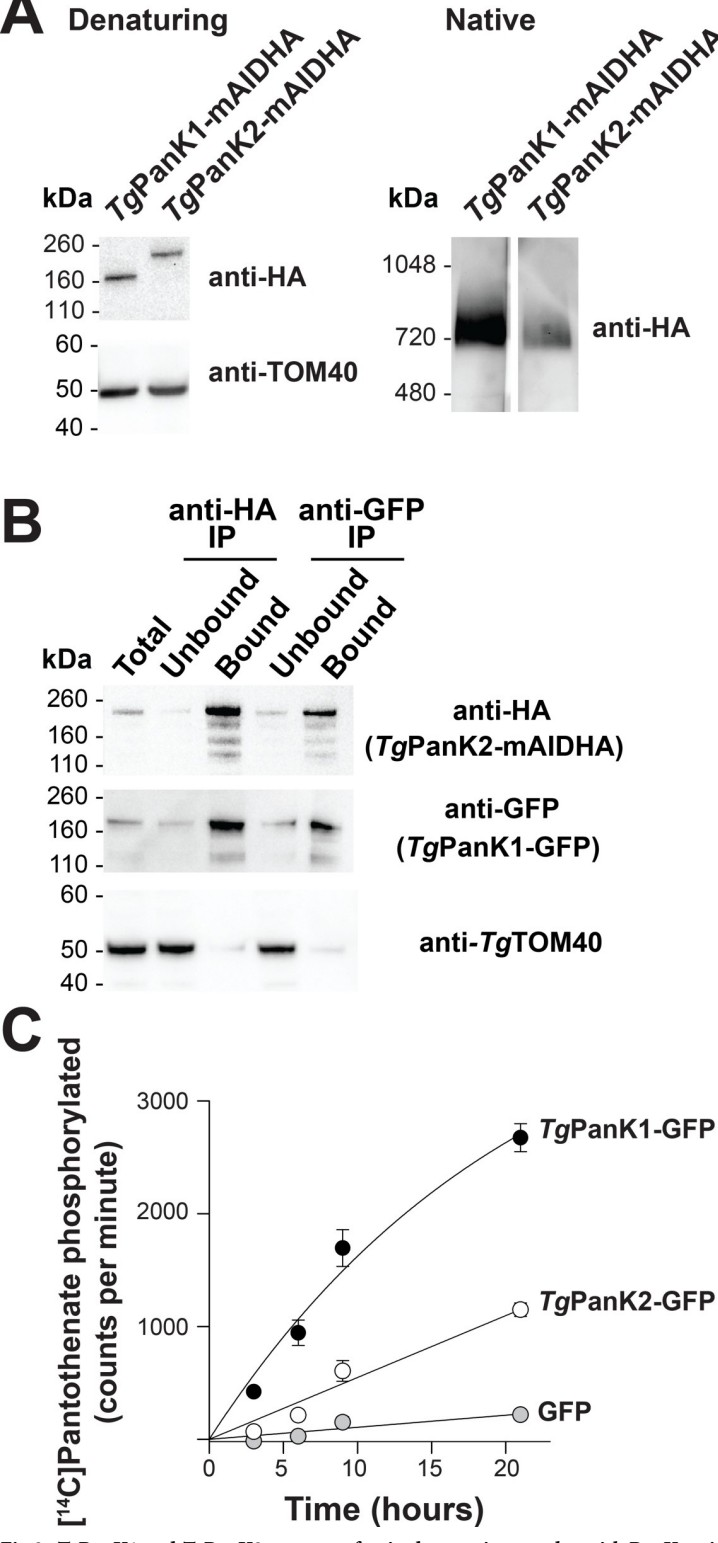

**Fig 3. *Tg*PanK1 and *Tg*PanK2 are part of a single protein complex with PanK activity.** (A) Denaturing and native western blot analyses of the HA-tagged proteins in *Tg*PanK1-mAIDHA and *Tg*PanK2-mAIDHA parasite lines. The expected sizes of *Tg*PanK1-mAIDHA and *Tg*PanK2-mAIDHA are ~143 kDa and ~189 kDa, respectively. Western blots were performed with an anti-HA antibody and each blot shown is a representative of three independent experiments, each performed with different batches of parasites. Denaturing western blots were also probed with anti-

*Tg*TOM40, which served as a loading control. (B) Western blot analysis of proteins from *Tg*PanK1-GFP/*Tg*PanK2-mAIDHA parasite lysates immunoprecipitated with GFP-Trap and anti-HA beads (*Tg*PanK1-GFP is 160 kDa). Protein samples were collected before immunoprecipitation (Total), from the fraction not bound to the GFP-Trap/anti-HA beads (Unbound), and from the fraction bound to the GFP-Trap/anti-HA beads (Bound). Membranes were probed with anti-GFP and anti-HA antibodies, and the blot shown is representative of three independent experiments, each performed with different batches of parasites. *Tg*TOM40 served as a control protein that is part of an unrelated protein complex. Bound fractions contain protein from 4 × as many cells as the total and unbound lanes. (C) The phosphorylation of [$^{14}$C]pantothenate (initial concentration 2 μM) over time by protein samples immunoprecipitated with GFP-Trap from *Tg*PanK1-GFP/*Tg*PanK2-mAIDHA (black circles),*Tg*PanK1-HA/*Tg*PanK2-GFP (white circles) and untagged GFP (grey circles) lines. Data shown are representative of two independent experiments, each performed with a different batch of parasites and carried out in duplicate. Error bars represent the range/2 and are not shown if smaller than the symbols.

PanK activity of the complex, we set out to complement the knocked down *Tg*PanKs with a well-characterised PanK. The *Staphylococcus aureus* PanK (*Sa*PanK) was selected for this purpose because (i) it is a type II PanK (like the *Tg*PanKs), (ii) it has been shown to function as a homodimer (so only the one gene needs to be expressed) and (iii) it is refractory to negative feedback inhibition by CoA (allowing the enzyme to function even if the parasite maintains high levels of CoA). We therefore constitutively expressed *Sapank* fused with the coding sequence for a Ty1 epitope tag in both the *Tg*PanK1-mAIDHA and *Tg*PanK2-mAIDHA parasite lines, generating lines that we termed *Tg*PanK1-mAIDHA$^{+Sa\text{PanK-Ty1}}$ and *Tg*PanK2-mAIDHA$^{+Sa\text{PanK-Ty1}}$. The expression of the *Sa*PanK protein in these strains was verified by immunofluorescence microscopy and western blot (**S11 Fig**). We measured the proliferation of the *Tg*PanK1-mAIDHA$^{+Sa\text{PanK-Ty1}}$ and *Tg*PanK2-mAIDHA$^{+Sa\text{PanK-Ty1}}$ lines in the presence and absence of 100 μM IAA, and compared this with the proliferation of the *Tg*PanK1-mAIDHA, *Tg*PanK2-mAIDHA and Parent lines. We obtained fluorescence measurements over a 7 day period and compared the proliferation of each strain when the Parent strain cultured in the absence of IAA reached mid-log phase. We found that both the *Tg*PanK1-mAIDHA$^{+Sa\text{PanK-Ty1}}$ and *Tg*PanK2-mAIDHA$^{+Sa\text{PanK-Ty1}}$ lines proliferated at a similar rate to the Parent control line when cultured in the presence of IAA, in contrast to the *Tg*PanK1-mAIDHA and *Tg*PanK2-mAIDHA lines, where minimal proliferation was observed (**Fig 4C**).

To test whether *Tg*PanK2 is required for *Tg*PanK1-dependent PanK activity in the parasites, we first investigated the stability of *Tg*PanK1 when *Tg*PanK2 was degraded by exposing the *Tg*PanK1-GFP/*Tg*PanK2-mAIDHA parasite line to 100 μM IAA over 6 hours. *Tg*PanK1 abundance remained unchanged following *Tg*PanK2 knockdown (**Fig 5A**), indicating that *Tg*PanK1 stability or turnover is not dependent on *Tg*PanK2. We next immunopurified *Tg*PanK1-GFP from *Tg*PanK1-GFP/*Tg*PanK2-mAIDHA parasites that were incubated in the presence (+IAA) or absence (-IAA) of 100 μM IAA for 2–3 h, and tested the purified protein for PanK activity (**S1 Fig**, **orange arrow**). Although we observed similar abundance of purified *Tg*PanK1-GFP between the +IAA and -IAA samples (**Fig 5B**), the samples immunoprecipitated from parasites exposed to IAA were devoid of PanK activity, whereas samples not exposed to IAA were able to phosphorylate pantothenate (**Fig 5C**). These data demonstrate that *Tg*PanK1 is inactive in the absence of *Tg*PanK2. Collectively, our studies on *Tg*PanK1 and *Tg*PanK2 reveal that (i) *Tg*PanK1 and *Tg*PanK2 are part of the same protein complex, (ii) expression of both proteins is required for PanK activity, and (iii) PanK activity of the complex is important for *T. gondii* proliferation during the disease-causing tachyzoite stage.

## Discussion

All PanKs characterised to date have been shown to exist as homodimers [3–10]. Here we present data consistent with PanK1 and PanK2 of the apicomplexan parasites *P. falciparum* and *T.*

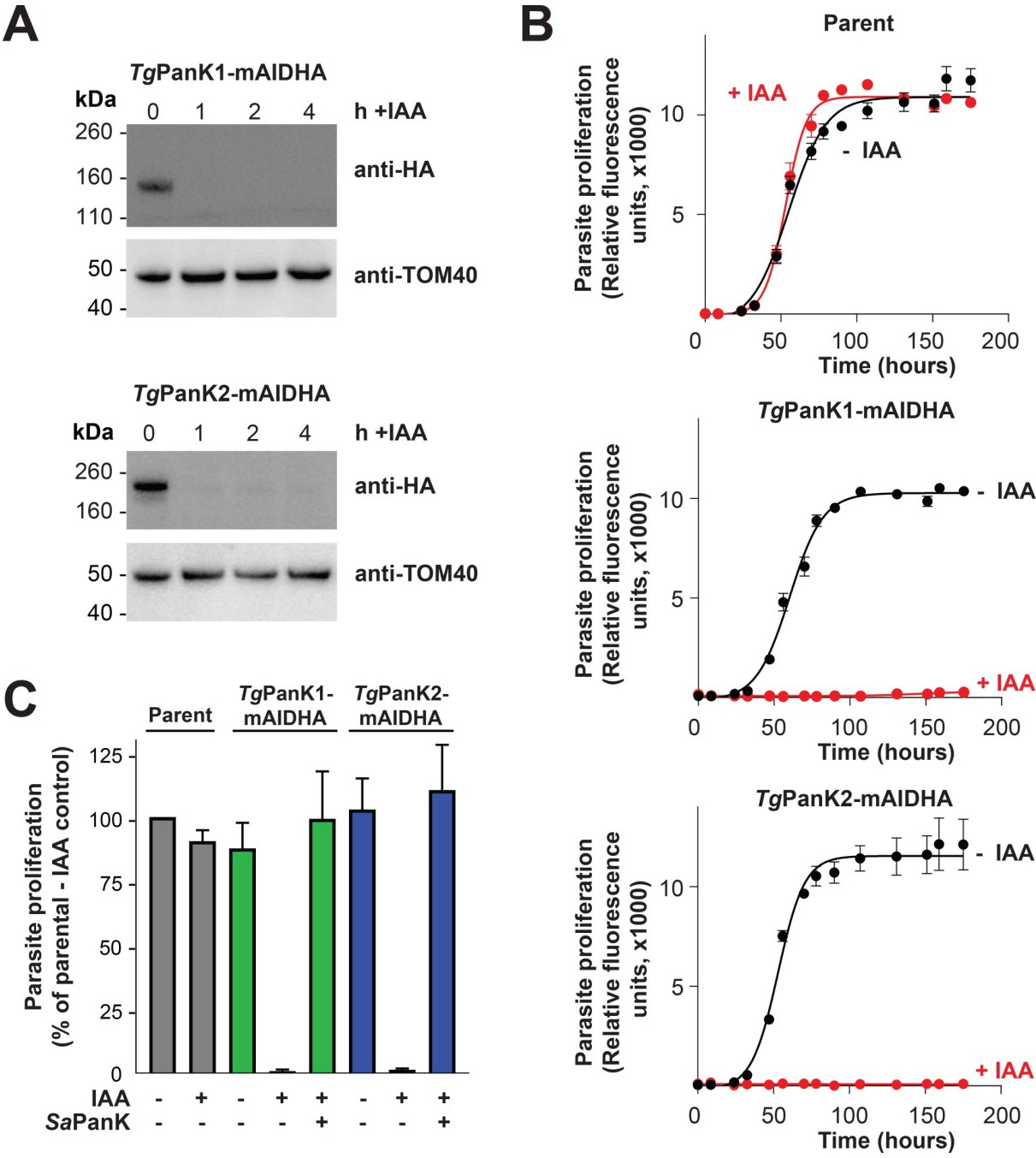

**Fig 4. Expression of both *Tg*PanK1 and *Tg*PanK2 is necessary for PanK activity and for *T. gondii* tachyzoite proliferation.** (A) IAA-induced knockdown of *Tg*PanK1-mAIDHA or *Tg*PanK2-mAIDHA protein over time. Western blot analysis of *Tg*PanK1-mAIDHA and *Tg*PanK2-mAIDHA lines incubated with either 100 μM IAA (+IAA) for 1, 2 and 4 h or an ethanol vehicle control (0 h). Membranes were probed with anti-HA antibody to detect the *Tg*PanK1-mAIDHA and *Tg*PanK2-mAIDHA proteins, and with anti-*Tg*Tom40 as a loading control. Western blots shown are representative of three independent experiments, each performed with a different batch of parasites. (B) The effect of *Tg*PanK1-mAIDHA or *Tg*PanK2-mAIDHA knockdown on *T. gondii* tachyzoite proliferation. Parent, *Tg*PanK1-mAIDHA and *Tg*PanK2-mAIDHA lines (all expressing tdTomato RFP) were cultured over 7 days in the presence (red circles) or absence (black circles) of 100 μM IAA. Parasite proliferation was measured over time by assessing the RFP expression using a fluorescence reader. Graphs shown are representative of three independent experiments carried out in triplicate, each performed with a different batch of parasites. Error bars represent SD and are not shown if smaller than the symbols. (C) Complementation of *Tg*PanK1 and *Tg*PanK2 knockdown with *Sa*PanK. *Sa*PanK was constitutively expressed (+) in *Tg*PanK1-mAIDHA[+*Sa*PanK-Ty1] (green bars) and *Tg*PanK2-mAIDHA[+*Sa*PanK-Ty1] (blue bars) parasites. These lines were cultured alongside the non-complemented (-) *Tg*PanK1-mAIDHA (green bars), *Tg*PanK2-mAIDHA (blue bars) and parent lines (grey bars). All parasite lines were cultured either in the presence (+) or absence (-) of 100 μM IAA. Parasite proliferation was

monitored 1–2 times daily for 7 days. Proliferation was compared when the Parent strain cultured in the absence of IAA was at the mid-log phase of parasite proliferation. Values are averaged from three independent experiments, each performed with a different batch of parasites and carried out in triplicate. Error bars represent SEM.

*gondii* forming a heteromeric complex (**Figs 1–3**), a hitherto undescribed phenomenon in nature.

There have been several attempts by us [unpublished] and others [37–41] to express a functional *Pf*PanK1. Whilst the protein has been successfully expressed in soluble form using various heterologous expression systems (*Escherichia coli*, insect cells, *Saccharomyces cerevisiae*), until recently [41], no study had reported PanK activity from the heterologously-expressed and purified protein. Nurkanto *et al.* [41] have recently reported the functional expression of *Pf*PanK1 in *E. coli*. The expressed protein was initially insoluble but was solubilised using high concentrations of trehalose. The authors characterised the protein's PanK activity (using an enzyme-coupled assay) and reported a pantothenate $K_M$ of 44.6 μM. This $K_M$ is more than two orders of magnitude higher than the $K_M$ that has been reported previously for *Pf*PanK activity in parasite lysates [27,42,43]. The physiological significance of the *Pf*PanK1 activity reported in the Nurkanto *et al.* study is therefore unclear. Our observations that the PanK activity of the immunoprecipitated *Pf*PanK complex (**Fig 2B(ii)**), which includes the presence of *Pf*PanK2 (as well as, potentially, additional proteins), is indistinguishable from the PanK activity of parasite lysates (**Fig 2B(i)**), is consistent with the *Pf*PanK complex described here being responsible for the malaria parasite's PanK activity.

A comparison of the amino acid sequences of *P. falciparum* and *T. gondii* PanKs with those of other type II PanKs, such as human PanK3 (*Hs*PanK3), provides a possible explanation for why PanKs from these apicomplexan parasites exist in heteromeric complexes (**S7 Fig**). Each of the two identical active sites of the homodimeric *Hs*PanK3 are formed by parts of both of its protomers. Certain residues form hydrogen bonds with pantothenate (Glu138, Ser195, Arg207 from one protomer and Val268' and Ala269' from the second protomer), while others interact to stabilise the active site (Asp137 with Tyr258', and Glu138 with Tyr254') [34,35] (**S12 Fig**). Notably, the hydrogen bond between Glu138 and Tyr254' is important for the allosteric activation of the enzyme [35]. Critically, one of the important residues involved in active site stabilisation, Asp137, is only conserved in the PanK1 of *P. falciparum* and *T. gondii* but not their PanK2, while others, such as Tyr254' and Tyr258' are conserved in their PanK2 but not PanK1 (**S7** and **S12 Figs**). This raises the possibility that PanK1 and PanK2 homodimers are not functional, and that only a heteromeric PanK1/PanK2 complex, with a single complete active site, can serve as a functional PanK enzyme in these apicomplexan parasites. This is consistent with the previous observation that two of the nucleotide-binding motifs of *Pf*PanK2 deviate from those of other eukaryotic PanKs [28]. Whether the incomplete second active site plays an additional, as yet undetermined, role(s) remains to be seen. It should be noted that the PanKs of other apicomplexan parasites (including species from the genera *Babesia*, *Cryptosporidium* and *Eimeria*) exhibit a similar conservation of residues as that described above for *P. falciparum* and *T. gondii* (**S13 Fig**), raising the possibility that heteromeric PanK complexes are ubiquitous in Apicomplexa.

The apparent molecular weight of the *Pf*PanK heterodimer complex (as determined from native western blotting) is consistent with that of a complex that includes *Pf*PanK1, *Pf*PanK2 and a *Pf*14-3-3I dimer (**Fig 1B**). However, due to various limitations of native gels [44], it is difficult to obtain an accurate estimate of the molecular weight of the complex. Although we cannot rule out the inclusion of other proteins in the *Pf*PanK complex, such as M17 leucyl aminopeptidase (**Fig 2A** and **S1 Table**), we think that this is unlikely, since peptides from these

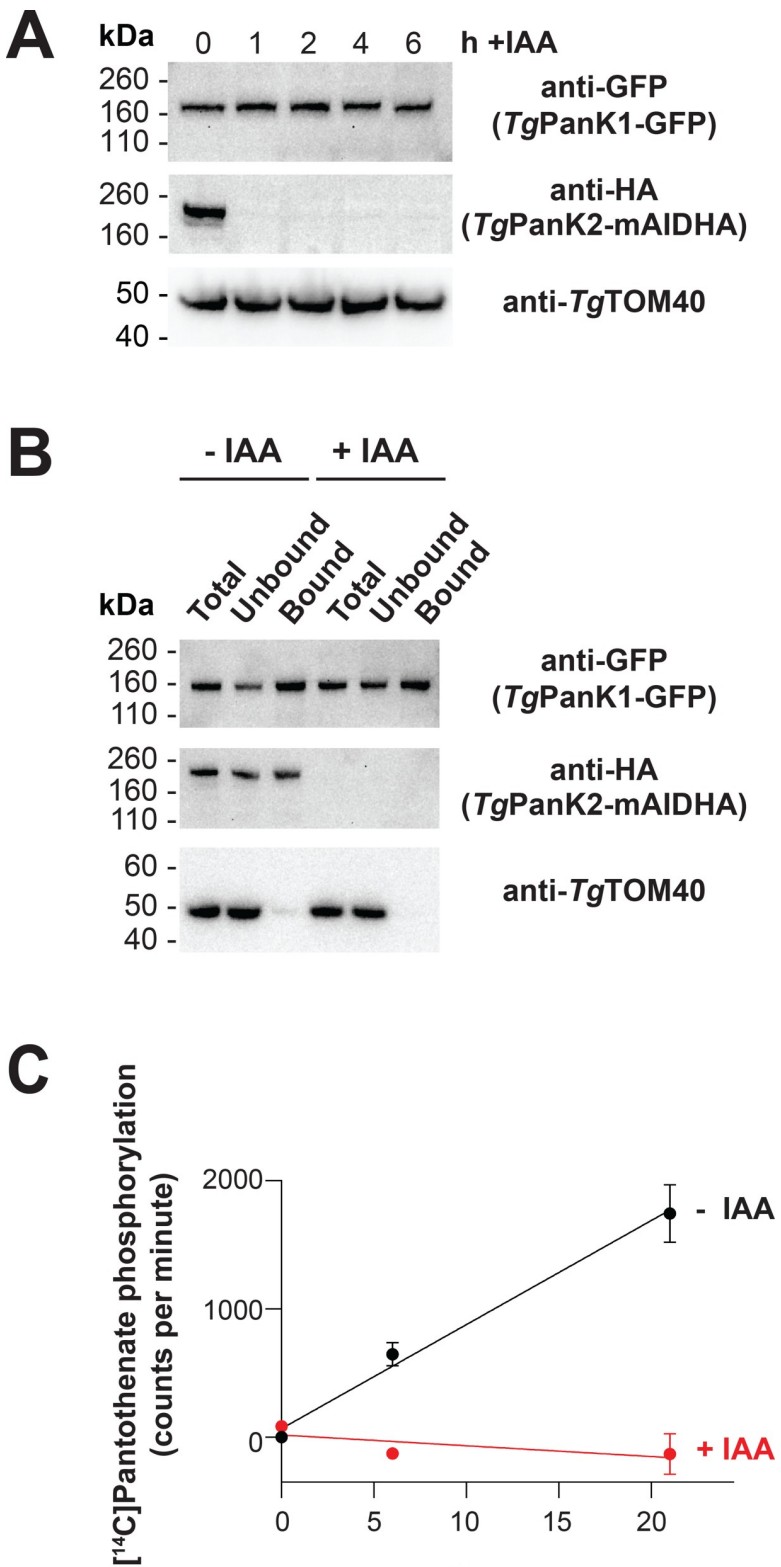

**Fig 5. *Tg*PanK2 is required for PanK activity in *T. gondii* parasites.** (A) Denaturing western blot analysis of the GFP- and HA-tagged proteins in the *Tg*PanK1-GFP/*Tg*PanK2-mAIDHA parasite line cultured in the absence or presence of IAA. *Tg*PanK1-GFP/*Tg*PanK2-mAIDHA parasites were incubated with either 100 μM IAA (+IAA) for 1,

2, 4 and 6 h, or an ethanol vehicle control (0 h). *Tg*TOM40 served as a loading control. (B) Denaturing western blot analysis of the GFP- and HA-tagged proteins in the *Tg*PanK1-GFP/*Tg*PanK2-mAIDHA parasite line before and after immunopurification. *Tg*PanK1-GFP/*Tg*PanK2-mAIDHA parasites were incubated with either 100 μM IAA (+IAA), or an ethanol vehicle control (-IAA) for 2–3 h. The cells were lysed and a sample of total lysate (Total) was incubated with GFP-Trap. The GFP-Trap-immunoprecipitated protein (Bound) samples were then separated from the supernatant (Unbound) and all three fractions analysed by western blot. *Tg*TOM40 served as a control protein that is not expected to be in the *Tg*PanK complex. The western blot shown is a representative of three independent experiments, each performed with a different batch of parasites. (C) The phosphorylation of [14C]pantothenate (initial concentration of 2 μM) over time by GFP-Trap-immunoprecipitated protein samples from *Tg*PanK1-GFP/*Tg*PanK2-mAIDHA parasite lysates. The lysates were generated from parasites that were incubated in either the presence (+IAA) or absence (-IAA) of 100 μM IAA for 2–3 h. Data shown are averaged from three independent experiments, each performed with a different batch of parasites and carried out in duplicate. Error bars represent SEM and are not shown if smaller than the symbols.

proteins were detected at lower abundance than peptides from *Pf*PanK1, *Pf*PanK2 and *Pf*14-3-3I. The role of *Pf*14-3-3I in the heteromeric *Pf*PanK complex (**Fig 2A and 2C**) is not clear. The 14-3-3 protein family comprises highly conserved proteins that occur in a wide array of eukaryotic organisms, including apicomplexans such as *P. falciparum* [32,45–47]. Multiple iso-forms of 14-3-3 are found to occur in every organism that expresses the protein [48]. 14-3-3 proteins bind to, and regulate, the function of proteins that are involved in a large range of cel-lular functions, including cell cycle regulation, signal transduction and apoptosis (reviewed in [49]). They typically bind to phosphorylated Ser/Thr residues on target proteins, and modify their target protein's trafficking/targeting (reviewed in [50]), conformation, co-localisation, and/or activity (reviewed in [51]). The *Tg*PanK heterodimer complex has a molecular weight that is much larger than the combined molecular weights of *Tg*PanK1 and *Tg*PanK2. Unfortu-nately, mass spectrometry analysis aimed at identifying the protein composition of the *T. gon-dii* PanK complex was unsuccessful, presumably because the native level of expression of the complex is too low. Nevertheless, we were unable to detect *Tg*14-3-3 in the *Tg*PanK complex with a pan-specific 14-3-3 antibody even though we could detect it in *T. gondii* parasite lysates (**S10 Fig**). Whether this means that the presence of 14-3-3 in the *P. falciparum* PanK complex is not essential for enzyme activity, or if a different protein fulfils a similar role to 14-3-3 in the *Tg*PanK complex, remains to be seen.

In this study, we have characterised, for the first time, PanK activity in *T. gondii*. The [14C]pantothenate phosphorylation data generated with the purified *Tg*PanK complex (**Fig 3C**) provide the first biochemical evidence indicating that these putative PanKs are able to phosphorylate pantothenate. This finding, combined with the results of the knockdown and *Sa*PanK complementation experiment in *T. gondii* (**Fig 4B** and **4C**), as well as our demonstra-tion that *Tg*PanK1 alone is inactive (**Fig 5**), not only demonstrate the essentiality of *Tg*PanK1 and *Tg*PanK2 and their dependence on each other, but also show that the essentiality is due to their role in phosphorylating pantothenate.

*T. gondii* parasites inhabit metabolically active mammalian cells that contain their own CoA biosynthesis pathway. Our data indicate that *T. gondii* parasites are unable to scavenge sufficient downstream intermediates in the CoA biosynthesis pathway, including CoA, from their host cells, for their survival, and therefore must maintain their own active CoA biosyn-thesis pathway. The requirement for CoA biosynthesis in *T. gondii*, coupled with the intense investigation of this pathway as a drug target in *P. falciparum* [27,39–41,52–67], suggests that further characterisation of *Tg*PanK, and the CoA biosynthesis pathway in *T. gondii*, could yield novel drug targets for chemotherapy.

It has been an open question as to why many organisms (eukaryotes [11–21] and prokary-otes [7,23–25]), including all apicomplexan parasites [68], express more than one PanK and why some PanKs appear to be non-functional [7,21,23] (either by analysis of their sequence or

through failed attempts to demonstrate PanK activity experimentally). The data that we present here provides a possible answer to this question, at least in apicomplexan parasites.

## Methods

### Parasite and host cell culture

*P. falciparum* parasites were maintained in RPMI 1640 medium supplemented with 11 mM glucose (to a final concentration of 22 mM), 200 μM hypoxanthine, 24 μg/mL gentamicin and 6 g/L Albumax II as described previously [69]. *T. gondii* was cultured in human foreskin fibroblasts (HFF cells) as described previously [70]. *T. gondii* parasites were grown in flasks with a confluent HFF cell layer in either Dulbecco's modified Eagle's medium (DMEM) or complete RPMI 1640, with both media containing 2 g/L sodium bicarbonate and supplemented with 1% (v/v) fetal bovine serum (FBS), 50 units/mL penicillin, 50 μg/mL streptomycin, 10 μg/mL gentamicin, 0.2 mM L-glutamine, and 0.25 μg/mL amphotericin B.

### Plasmid preparation and parasite transfection

The *Pf*PanK1-GFP line was generated in a previous study [27], while the untagged GFP line was a generous gift from Professor Alex Maier (Research School of Biology, Australian National University, Canberra). A *Pfpank2*-pGlux-1 vector was generated for the overexpression of *Pf*PanK2-GFP in 3D7 strain *P. falciparum* as detailed in **S1 Text**. The primers used are listed in **S2 Table**. The same construct was also transfected into each of the *Pf*PanK1 mutants and their Parent line described previously by Tjhin *et al.* [27]. Transfections were performed with ring-stage parasites and transformants were subsequently selected and maintained using WR99210 (10 nM) as described previously [71].

Transgenic *T. gondii* parasite lines were generated using a CRISPR/Cas9 strategy as previously described in Shen *et al.* [72], which is detailed in the **S1 Text**. The guide RNAs, primers, and the sequences of gBlocks used are provided in **S2** and **S3 Tables.**

The complementation lines *Tg*PanK1-mAIDHA$^{+Sa\text{PanK-Ty1}}$ and *Tg*PanK2-mAIDHA$^{+Sa\text{-PanK-Ty1}}$ were created by expressing the *S. aureus* type II PanK gene (*Sapank*) in *T. gondii* under the regulation of the tubulin promoter (details in the **S1 Text, S2** and **S3 Tables).**

### Immunofluorescence assays and microscopy

Fixed *Pf*PanK2-GFP-expressing 3D7 strain *P. falciparum* parasites within infected erythrocytes were observed and imaged with a Leica TCS-SP2-UV confocal microscope (Leica Microsystems) using a 63× water immersion lens as described in the **S1 Text**. To confirm the expression of *Sa*PanK-Ty1 in the *Tg*PanK1-mAIDHA$^{+Sa\text{PanK-Ty1}}$ line, immunofluorescence assays were performed based on the protocol described by van Dooren *et al.* [73]. *T. gondii* parasites were incubated with mouse anti-Ty1 antibodies (1:200 dilution). Secondary antibodies used were goat anti-mouse AlexaFluor 488 at a 1:250 dilution. The nucleus was stained with DAPI. Immunofluorescence images were acquired on a DeltaVision Elite system (GE Healthcare) using an inverted Olympus IX71 microscope with a 100× UPlanSApo oil immersion lens (Olympus) paired with a Photometrics CoolSNAP HQ$^2$ camera. Images taken on the DeltaVision setup were deconvolved using SoftWoRx Suite 2.0 software. Images were adjusted linearly for contrast and brightness.

### Polyacrylamide gel electrophoresis and western blotting

Parasite protein samples were analysed using either denaturing or blue native gels to determine the presence and abundance of a single protein or protein complex of interest,

respectively. Briefly, mature trophozoite-stage *P. falciparum* parasites were isolated from infected erythrocytes by saponin lysis, as described previously [74]. Saponin-isolated parasites were then pelleted and lysed in the appropriate buffers (as detailed in the **S1 Text**). *T. gondii* protein samples were prepared as described previously, with samples for blue native-PAGE solubilised in Native PAGE sample buffer (ThermoFisher) containing 1% (v/v) Triton X-100 [73]. Protein samples generated from both *P. falciparum* and *T. gondii* parasites were separated by polyacrylamide gel electrophoresis (PAGE) in precast NuPAGE (4–12% or 12%) or Native-PAGE (4–16%) gels (ThermoFisher) according to the manufacturer's instructions with minor modifications (detailed in the **S1 Text**). The separated proteins were transferred to the appropriate membranes (nitrocellulose or polyvinylidene fluoride (PVDF)) and blocked (detailed in the **S1 Text**) before immunoblotting. Blocked membranes were exposed (45 min– 2 h) to specific primary and secondary antibodies to allow for the detection of the protein of interest. To visualise the protein band(s), membranes were incubated in Pierce enhanced chemiluminescence (ECL) Plus Substrate (ThermoFisher) according to the manufacturer's instructions or home-made ECL substrate (0.04% w/v luminol, 0.007% w/v coumaric acid, 0.01% v/v $H_2O_2$, 100 mM Tris, pH 9.35). Protein bands were then either imaged onto X-ray films and scanned or visualised on a ChemiDoc MP Imaging System (Bio-Rad).

## Flow cytometry

Saponin-isolated mature trophozoites from 3D7 wild-type, Parent$^{+Pf\text{PanK2-GFP}}$, PanOH-A$^{+Pf\text{PanK2-GFP}}$, PanOH-B$^{+Pf\text{PanK2-GFP}}$ and CJ-A$^{+Pf\text{PanK2-GFP}}$ cultures were subjected to flow cytometry analysis to determine the proportion of GFP-positive cells (**S6 Fig**). Aliquots of each isolated parasite suspension were diluted in a saline solution (125 mM NaCl, 5 mM KCl, 25 mM HEPES, 20 mM glucose and 1 mM $MgCl_2$, pH 7.1) to a concentration of $\sim 10^6$–$10^7$ cells/mL in 1.2 mL Costar polypropylene cluster tubes (Corning) and sampled for flow cytometry analysis (in measurements of 100,000 cells, low sampling speed) with the following settings: forward scatter = 450 V (log scale), side scatter = 350 V (log scale) and AlexaFluor 488 = 600 V (log scale). The 3D7 wild-type cells were used to establish a gating strategy that defined a threshold below which parasites were deemed to be auto-fluorescent. This strategy was then applied in all analyses to determine the proportion of cells in each cell line that was GFP-positive (i.e. above the defined threshold).

## Immunoprecipitations

In order to immunopurify GFP-tagged or HA-tagged proteins from parasite lysates, immunoprecipitation was performed using either GFP-Trap (high affinity anti-GFP alpaca nanobody bound to agarose beads; Chromotek) or anti-HA beads (Sigma-Aldrich), respectively. *P. falciparum* lysate was prepared from saponin-isolated trophozoites, and *T. gondii* lysate was prepared from tachyzoites, as described previously ([74] and [73], respectively). Immunoprecipitation was then performed (as detailed in the **S1 Text**). In *P. falciparum* experiments where the amount of immunoprecipitated proteins were to be standardised across cell lines and biological repeats, the number of GFP-positive cells to be used for lysate preparation was calculated by a combination of haemocytometer count and flow cytometry. All immunoprecipitated samples from Parent$^{+Pf\text{PanK2-GFP}}$, PanOH-A$^{+Pf\text{PanK2-GFP}}$, PanOH-B$^{+Pf\text{PanK2-GFP}}$ and CJ-A$^{+Pf\text{PanK2-GFP}}$ cell lines contained protein from $5 \times 10^7$ GFP-positive cells. Each of these samples was subsequently divided into two equal aliquots, one used in the [$^{14}$C]pantothenate phosphorylation assay and the other for denaturing western blot.

When an aliquot of the immunoprecipitation sample (beads that have bound proteins from $\sim 10^6$–$10^7$ GFP-positive cells for *P. falciparum* and $\sim 10^7$–$10^8$ cells for *T. gondii*) was required

for western blot, the bead suspension was centrifuged (2,500 × g, 2 min), the supernatant removed, and the beads resuspended in 50 μL sample buffer containing 2 × NuPAGE lithium dodecyl sulfate (LDS) sample buffer (ThermoFisher) and 2 × NuPAGE sample reducing agent (ThermoFisher). In some experiments, 10 μL aliquots of the total or unbound lysate fractions were each mixed with 10 μL of the same sample buffer. These samples were then boiled (95˚C, 10 min) and 10 μL of each was then used in a denaturing western blot as described above.

## [$^{14}$C]Pantothenate phosphorylation assays

In order to determine the PanK activity of the protein(s) isolated in the GFP-Trap immunoprecipitation assays, the immunopurified complexes were used to perform a [$^{14}$C]pantothenate phosphorylation time course. The bead suspensions containing the immunoprecipitated proteins from *P. falciparum* and *T. gondii* were centrifuged (2,500 × g, 2 min), the supernatant removed, and the beads resuspended in 250–300 μL (*T. gondii*) or 500 μL (for *P. falciparum*) buffer containing 100 mM Tris-HCl (pH 7.4), 10 mM ATP and 10 mM MgCl$_2$ (i.e. all reagents were at twice the final concentration required for the phosphorylation reaction). Each time course was then initiated by the addition of 250–300 μL (for *T. gondii*) or 500 μL (for *P. falciparum*) 4 μM (0.2 μCi/mL) [$^{14}$C]pantothenate in water (pre-warmed to 37˚C), to the bead suspension. Aliquots of each reaction (50 μL in duplicate) were terminated at pre-determined time points by mixing with 50 μL 150 mM barium hydroxide preloaded within the wells of a 96-well, 0.2 μm hydrophilic PVDF membrane filter bottom plate (Corning). Phosphorylated compounds in each well were then precipitated by the addition of 50 μL 150 mM zinc sulfate to generate the Somogyi reagent [75], the wells processed, and the radioactivity in the plate determined as detailed previously [43]. Total radioactivity in each phosphorylation reaction was determined by mixing 50 μL aliquots of each reaction (in duplicate) thoroughly with 150 μL Microscint-40 (PerkinElmer) by pipetting the mixture at least 5 times, in the wells of an OptiPlate-96 microplate (PerkinElmer) [43].

## Mass spectrometry of immunoprecipitated samples

The identities of the proteins co-immunoprecipitated from lysates of the wild-type 3D7, *Pf*PanK1-GFP, *Pf*PanK2-GFP and untagged GFP lines were determined by mass spectrometry. Aliquots of bead-bound co-immunoprecipitated samples were resuspended in 2 × NuPAGE LDS sample buffer and 2 × NuPAGE sample reducing agent and sent (at ambient temperature, travel time less than 24 h) to the Australian Proteomics Analysis Facility (Sydney) for processing and mass spectrometry analysis (as detailed in the **S1 Text**).

## Fluorescent *T. gondii* proliferation assay

Fluorescent *T. gondii* proliferation assays were performed as previously described [36]. Briefly, 2000 parasites suspended in complete RPMI were added to the wells of optical bottom black 96 well plates (ThermoFisher) containing a confluent layer of HFF cells, either in the presence of 100 μM IAA dissolved in ethanol (final ethanol concentration of 0.1%, v/v) or with ethanol (0.1%, v/v) as a vehicle control, in triplicate. Fluorescent measurements (Excitation filter, 540 nm; Emission filter, 590 nm) using a FLUOstar OPTIMA Microplate Reader (BMG LABTECH) were taken over 7 days.

## Knockdown of mAID protein

Flasks containing a confluent layer of HFF cells were seeded with *Tg*PanK1-mAIDHA, *Tg*PanK2-mAIDHA, *Tg*PanK1-mAIDHA$^{+SaPanK-Ty1}$ or *Tg*PanK2-mAIDHA$^{+SaPanK-Ty1}$

*T. gondii* parasites. While the parasites were still intracellular, 100 μM of IAA dissolved in ethanol (final ethanol concentration of 0.1%, v/v) was added to induce the knockdown of *Tg*PanK1-mAIDHA or *Tg*PanK2-mAIDHA, with ethanol (0.1%, v/v) added to another flask as a vehicle control. Flasks with IAA added were processed at 1, 2 and 4 h time points, and the control flask was processed at the 4-h time point. Parasite concentrations were determined using a haemocytometer and $1.5 \times 10^7$ parasites were resuspended in LDS sample buffer, and boiled at 95˚C for 10 minutes. An aliquot from each sample was analysed by western blotting. The knockdown of *Tg*PanK2-mAIDHA protein in the *Tg*PanK1-GFP/*Tg*PanK2-mAIDHA line was carried out using the same protocol, but with the addition of a 6 h time point.

### Alignment of PanKs

PanK homologues from *P. falciparum* and *T. gondii*, and a selection of other type II PanKs from other eukaryotic organisms and *S. aureus* were aligned using PROMALS3D [76] (available at: http://prodata.swmed.edu/promals3d/promals3d.php). The default parameters were selected except for the 'Identity threshold above which fast alignment is applied' parameter, which was changed to "1" to allow for a more accurate alignment. The following PanK type II homologues were aligned (accession number included in brackets): *Staphylococcus aureus* (Q2FWC7); *Saccharomyces cerevisiae* (Q04430); *Aspergillus nidulans* (O93921); *Homo sapiens* PanK1 (Q8TE04), PanK2 (Q9BZ23), PanK3 (Q9H999) and PanK4 (Q9NVE7); *Arabidopsis thaliana* PanK1 (O80765) and PanK2 (Q8L5Y9); *Plasmodium falciparum* PanK1 (Q8ILP4) and PanK2 (Q8IL92) and *Toxoplasma gondii* PanK1 (A0A125YTW9) and PanK2 (V5B595).

### Statistical analysis

Statistical analysis between the means of the pantothenate phosphorylation rate by the Parent lysate and that of the immunopurified complex from the Parent$^{+Pf\text{PanK2-GFP}}$ line was carried out with unpaired, two-tailed, Student's t test using GraphPad Prism 8 (GraphPad Software, Inc) from which the 95% confidence interval of the difference between the means (95% CI) was obtained. All regression analysis was done using SigmaPlot version 11.0 for Windows (Systat Software, Inc) or GraphPad Prism 8 (GraphPad Software, Inc).

## Supporting information

**S1 Table. List of proteins identified in the MS analysis of the GFP-Trap immunoprecipitated complexes from *Pf*PanK1-GFP and *Pf*PanK2-GFP lines.** Proteins detected in each immunoprecipitation experiment are listed in descending order according of the total number of peptides detected across the two replicates (total peptides in 1$^{st}$ and 2$^{nd}$ rep columns). Only proteins that are present in the immunoprecipitation fractions of both parasite lines and absent in the negative controls (bound fractions of untagged GFP and 3D7 parasite lysates) are shown. Proteins shown in Fig 2A are indicated in red.
(TIF)

**S2 Table. List of oligonucleotides used in this study.**
(TIF)

**S3 Table. List of gBLOCK sequences used in this study.**
(TIF)

**S1 Fig. Overview of experimental workflow.** Flow chart highlighting the cell lines generated for the study and the main experimental steps that were performed. The coloured arrows represent final experimental results, and the associated figures within which the data are

presented, are indicated under each experiment.
(TIF)

**S2 Fig. GFP-Trap immunoprecipitation of proteins from *Pf*PanK1-GFP, *Pf*PanK2-GFP and untagged GFP lines.** Denaturing western blot analysis of the GFP-tagged proteins present in the total lysate, unbound and GFP-Trap-bound fractions of *Pf*PanK1-GFP, *Pf*PanK2-GFP and untagged GFP lines. Western blots were performed with anti-GFP antibody and the blot shown is representative of two independent experiments each performed with a different batch of parasites.
(TIF)

**S3 Fig. MS coverage of *Pf*PanK1.** *Pf*PanK1 peptides detected in the two independent MS analyses of the GFP-Trap immunoprecipitation from the *Pf*PanK1-GFP and *Pf*PanK2-GFP lines. Residues in green were detected in either analysis with >95% confidence, while residues in orange were detected in either analysis with >90% (but <95%) confidence. Percentage coverage was calculated using only the residues labelled green.
(TIF)

**S4 Fig. MS coverage of *Pf*PanK2.** *Pf*PanK2 peptides detected in the two independent MS analyses of the GFP-Trap immunoprecipitation from the *Pf*PanK1-GFP and *Pf*PanK2-GFP lines. Residues in green were detected in either analysis with >95% confidence, while residues in orange were detected in either analysis with >90% (but <95%) confidence. Percentage coverage was calculated using only the residues labelled green.
(TIF)

**S5 Fig. MS coverage of *Pf*14-3-3I.** *Pf*14-3-3I peptides detected in the two independent MS analyses of the GFP-Trap immunoprecipitation from the *Pf*PanK1-GFP and *Pf*PanK2-GFP lines. Residues in green were detected in either analysis with >95% confidence, while residues in orange were detected in either analysis with >90% (but <95%) confidence. Percentage coverage was calculated using only the residues labelled green.
(TIF)

**S6 Fig. Determining the amount of GFP-Trap bound *Pf*PanK2-GFP for pantothenate phosphorylation assays.** (A) The proportion of GFP-positive saponin-isolated 3D7, Parent$^{+Pf\text{-}PanK2\text{-}GFP}$, PanOH-A$^{+Pf\text{PanK2-GFP}}$, PanOH-B$^{+Pf\text{PanK2-GFP}}$ and CJ-A$^{+Pf\text{PanK2-GFP}}$ trophozoites was determined by FACS analysis. The forward scatter (FSC) intensity on each x-axis corresponds to cell size and the y-axis corresponds to the intensity of GFP fluorescence. The proportion of GFP-positive cells in each transgenic line (percentage value in each plot) was determined by using 3D7 trophozoites to set a gating threshold below which parasites were defined to be auto-fluorescent. Data shown are representative of three independent experiments, each performed prior to the [$^{14}$C]pantothenate phosphorylation assays presented in Fig 2B(ii). The flow cytometry data were used to standardise the amount of *Pf*PanK2-GFP immunoprecipitated from each cell line used in each [$^{14}$C]pantothenate phosphorylation assay. (B) Denaturing western blot analysis of *Pf*PanK2-GFP in the GFP-Trap immunoprecipitated complexes that were used in the [$^{14}$C]pantothenate phosphorylation assays performed to generate the data in Fig 2B(ii). Western blots were performed with an anti-GFP antibody and each blot shows the relative amounts of *Pf*PanK2-GFP immunopurified from the four different cell lines used in each of the three [$^{14}$C]pantothenate phosphorylation experiment. The same volume of samples (10 μL per lane) was used for all three experiments.
(TIF)

**S7 Fig. Multiple sequence alignment of representative Type II PanKs.** The conserved PHOSPHATE 1, PHOSPHATE 2, and ADENOSINE 1 motifs of the acetate and sugar kinases/ Hsc70/actin (ASKHA) superfamily of kinases are labelled at the top of the alignment. The Glu (E) residue involved in catalysis and the Arg (R) residue involved in positioning the substrate, are shown on a black background. Residues that have been found to interact with pantothenate and acetyl-CoA in human PanK3 [34,35] are marked with a blue asterisk. Residues that were found to interact to stabilise the human PanK3 active site are marked with a red asterisk. The catalytic Glu (E) residue is marked with a red and blue asterisk as it is involved in both the interaction with pantothenate and the stabilisation of the active site through interaction with a Tyr (Y) residue of the opposite protomer. The numbers at the start and end of each sequence indicate the position of the first and last residue in the alignment, respectively. The lengths of insertions are specified within the square brackets and the total length of protein sequences are shown in round brackets. Residues within the ASKHA superfamily motifs and conserved residues are highlighted based on the consensus AA guide for the column as follows: identical = bold, hydrophobic (W,F,Y,M,L,I,V,A,C,T,H) = yellow, charged/polar/small (D,E,K, R,H/D,E,H,K,N,Q,R,S,T/A,G,C,S,V,N,D,T,P) = grey and Gly (G) = red. The two insertion regions (Ins 1 and Ins 2) common to eukaryotic type II PanKs, but absent in prokaryotic PanKs are indicated by the black horizontal bars, while the *Pf*Pank1/*Tg*PanK1 and *Pf*PanK2/ *Tg*PanK2 specific inserts are highlighted on a red and blue background, respectively. Conservation refers to the conservation index [77]. Values at and above the conservation index cutoff (5) are displayed above the amino acid. Consensus AA: refers to the consensus level alignment parameters for the consensus amino acid sequence. This is displayed if the weighted frequency of a certain class of residues in a position is above 0.8. Consensus symbols: conserved amino acids are in bold and uppercase letters; aliphatic (I, V, L): *l*; aromatic (Y, H, W, F): @; hydrophobic (W, F, Y, M, L, I, V, A, C, T, H): *h*; alcohol (S, T): o; polar residues (D, E, H, K, N, Q, R, S, T): p; tiny (A, G, C, S): t; small (A, G, C, S, V, N, D, T, P): s; bulky residues (E, F, I, K, L, M, Q, R, W, Y): b; positively charged (K, R, H): +; negatively charged (D, E): -; charged (D, E, K, R, H): c. Marked below the alignment, 85% consensus includes those residues that occur in either the superfamily motifs and/or conserved residues where the same residue occurs more than 85% (10 out of 13 sequences). Consensus secondary structure (ss) elements: h = alpha helix, e = beta strand. Species names are abbreviated as follows: *Sa = Staphylococcus aureus*, *Sc = Saccharomyces cerevisiae*, *An = Aspergillus nidulans*, *Hs = Homo sapiens*, *At = Arabidopsis thaliana*, *Pf = Plasmodium falciparum* and *Tg = Toxoplasma gondii*. The alignment was created using PROMALS3D [76].
(TIF)

**S8 Fig. Schematic of the native and modified *pank* genomic loci in *T. gondii* and verification of the incorporation of the coding sequence for various epitope tags into the *Tgpank1* and *Tgpank2* loci.** (A) Schematic of the *Tgpank1* and *Tgpank2* genomic loci, indicating the incorporation sites of the epitope tag coding sequence. The expected sizes of the PCR products when screened with each set of screening primers are shown above the corresponding epitope tag coding sequence. The screening primers are *Tgpank1* screen fwd and rvs for *Tgpank1* (referred to as *Tgpank1* primers in panels B-D), and *Tgpank2* screen fwd and rvs for *Tgpank2* (referred to as *Tgpank2* primers in panels B-D). Primers are detailed in S2 Table. (B) PCR analysis of the parental strain (*Tg*Parent), and singly-tagged *Tg*PanK1-mAIDHA and *Tg*PanK2-mAIDHA lines. Both *Tg*PanK1-mAIDHA and *Tg*PanK2-mAIDHA have successfully incorporated mAIDHA tags. (C) PCR analysis of the doubly-tagged *Tg*PanK1-GFP/*Tg*PanK2-mAIDHA line. CRISPR/Cas9 was utilised to incorporate a sequence encoding a TEV-GFP tag into the genomic locus of the *Tgpank1* gene within the *Tg*PanK2-mAIDHA line. (D) PCR

analysis of the *Tg*PanK1-HA/*Tg*PanK2-GFP doubly-tagged line. CRISPR/Cas9 was utilised to incorporate a sequence encoding a TEV-HA tag into the genomic locus of the *Tgpank1* gene and a sequence encoding a TEV-GFP tag into the genomic locus of the *Tgpank2* gene.
(TIF)

**S9 Fig. Anti-GFP and anti-HA immunoprecipitation of *Tg*PanK1-HA/*Tg*PanK2-GFP expressing parasites.** Anti-HA and anti-GFP denaturing western blot analysis of fractions from GFP-Trap and anti-HA immunoprecipitations performed using lysates prepared from the parasite lines expressing *Tg*PanK1-HA/*Tg*PanK2-GFP. The expected molecular masses of *Tg*PanK1-HA and *Tg*PanK2-GFP are 136 kDa and 206 kDa, respectively. The blot shown is representative of three independent experiments, each performed with a different batch of parasites. Denaturing western blots were also probed with anti-*Tg*TOM40, which served as a control for a protein that is not part of the PanK complex.
(TIF)

**S10 Fig. Absence of 14-3-3 from the *Tg*PanK heteromeric complex.** Anti-HA, anti-GFP and anti-14-3-3 denaturing western blot analysis of fractions from GFP-Trap immunoprecipitation of lysates prepared from the parasite line expressing *Tg*PanK1-GFP/*Tg*PanK2-mAIDHA. The expected molecular masses of *Tg*PanK1-GFP, *Tg*PanK2-mAIDHA and 14-3-3 are approximately 160 kDa, 189 kDa and 37 kDa, respectively. The blots shown are representative of two independent experiments, each performed with a different batch of parasites.
(TIF)

**S11 Fig. Expression of *Sa*PanK-Ty1 in parasites expressing *Tg*PanK1-mAIDHA and *Tg*PanK2-mAIDHA.** (A) Anti-HA and anti-Ty1 denaturing western blot analysis of *Sa*PanK-Ty1-complemented and non-complemented (i) *Tg*PanK1-mAIDHA and (ii) *Tg*PanK2-mAIDHA lines, in the absence or presence (for 1 h) of 100 μM IAA. The expected molecular masses of *Tg*PanK1-mAIDHA, *Tg*PanK2-mAIDHA and *Sa*PanK-Ty1 are ~143 kDa, ~189 kDa and ~29 kDa, respectively. Denaturing western blots were also probed with anti-*Tg*TOM40, which served as a loading control. Each blot shown is representative of three independent experiments, each performed with a different batch of parasites. (B) Fluorescence micrographs of a HFF cell infected with four tachyzoite-stage *Tg*PanK1-mAIDHA[+*Sa*PanK-Ty1] parasites within a vacuole, indicating the presence of *Sa*PanK-Ty1. From left to right: Differential interference contrast (DIC), tdTomato RFP (a marker of the nucleus and cytosol; red), anti *Sa*PanK-Ty1 (green), and merged images. Scale bar represents 2 μm.
(TIF)

**S12 Fig. Pantothenate binding site and interactions in *H. sapiens* PanK3.** (A) *H. sapiens* AMP-PNP-pantothenate-bound PanK3 crystal structure (PDB ID: 5KPR, Subramanian *et al*. [35]). The homodimeric protein is made up of two identical protomers (lilac and yellow) forming two identical active sites, each binding pantothenate (carbon atoms shown in green). The red square encompasses one of the active sites. (B) Magnification of the region outlined by the red square in (A). Residues from both protomers contribute to the stabilisation of the binding pocket (E138 forms a hydrogen bond with Y254' and D137 with Y258') and interact with pantothenate (E138, S195, R207, A269' and V268'). Hydrogen bonds with and between the sidechains of these residues are shown in red. An apostrophe denotes residues from the lilac protomer. (C) List of residues annotated in the *Hs*PanK3 model that participate in the stabilisation of the binding pocket (highlighted cyan), and a comparison to the equivalent residues in *P. falciparum* and *T. gondii* PanKs. The PanKs from *P. falciparum* and *T. gondii* do not individually contain the complete set of residues required for the stabilisation of the binding pocket, but the combination of residues (highlighted cyan) from PanK1 and PanK2 suggests

that each PanK1/PanK2 heterodimer will have only one stabilised binding site.
(TIF)

**S13 Fig. Multiple sequence alignment of active site stabilisation residues in apicomplexan PanKs.** The apicomplexan PanK residues corresponding to the *Hs*PanK3 residues that are involved in the stabilisation of the binding pocket (D137, E138, Y254 and Y258) are highlighted in cyan if they are conserved and in grey if they are not conserved. The numbers before each alignment indicate the position of the first residue in the alignment. Each apicomplexan PanK is grouped into PanK1 or PanK2 based on their similarity to either *Pf*PanK1/*Tg*PanK1 or *Pf*PanK2/*Tg*PanK2, respectively. The alignment was created using PROMALS3D [76].
(TIF)

**S1 Text. Supplementary Methods.**
(DOCX)

## Acknowledgments

Proteomics was undertaken at APAF, the infrastructure provided by the Australian Government through the National Collaborative Research Infrastructure Strategy (NCRIS). We are grateful to the Canberra Branch of the Australian Red Cross Lifeblood for the provision of red blood cells, and Professor Alex Maier (ANU) for the untagged GFP-expressing *P. falciparum* parasites and pGlux-1 plasmid.

## Author Contributions

**Conceptualization:** Erick T. Tjhin, Vanessa M. Howieson, Christina Spry, Giel G. van Dooren, Kevin J. Saliba.

**Data curation:** Erick T. Tjhin, Vanessa M. Howieson, Christina Spry.

**Formal analysis:** Erick T. Tjhin, Vanessa M. Howieson, Christina Spry.

**Funding acquisition:** Giel G. van Dooren, Kevin J. Saliba.

**Investigation:** Erick T. Tjhin, Vanessa M. Howieson, Christina Spry.

**Methodology:** Erick T. Tjhin, Vanessa M. Howieson, Christina Spry, Giel G. van Dooren, Kevin J. Saliba.

**Project administration:** Giel G. van Dooren, Kevin J. Saliba.

**Resources:** Giel G. van Dooren.

**Supervision:** Giel G. van Dooren, Kevin J. Saliba.

**Validation:** Erick T. Tjhin, Vanessa M. Howieson, Christina Spry.

**Visualization:** Erick T. Tjhin, Vanessa M. Howieson, Christina Spry, Kevin J. Saliba.

**Writing – original draft:** Erick T. Tjhin, Vanessa M. Howieson.

**Writing – review & editing:** Erick T. Tjhin, Vanessa M. Howieson, Christina Spry, Giel G. van Dooren, Kevin J. Saliba.

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
