## [Decision Letter · Decision Letter 0]

10 Mar 2021

Dear Professor Saliba,

Thank you very much for submitting your manuscript "A novel heteromeric pantothenate kinase complex in apicomplexan parasites" for consideration at PLOS Pathogens. As with all papers reviewed by the journal, your manuscript was reviewed by members of the editorial board and by several independent reviewers. In light of the reviews (below this email), we would like to invite the resubmission of a significantly-revised version that takes into account the reviewers' comments.

As you will see from the comments, all of the reviewers appreciate the work, the quality of the data and the contribution to the field. Reviewers 1, 2 and 3 suggest making relatively minor revisions, most of which are aimed at enhancing and clarifying different aspects of the results. Reviewer 4 made several suggestions and brought to your attention an article that was published today concerning PfPanK1 activity. In your response to the suggestions made by Reviewer 4, please pay specific attention to these points:

- Provide specific activity units in Figure 1c (as done in Fig 2).

- Determine the PanK activity of material (IP or lysate) from the Tg line TgPanK1-GFP/TgPanK2-mAIDHA after inducing knockdown of PanK2.

- Provide information about whether Tg14-3-3I can be detected with pan anti-14-3-3 in the TgPanK complex. Reviewer 1 also brought up this point.

We cannot make any decision about publication until we have seen the revised manuscript and your response to the reviewers' comments. Your revised manuscript is also likely to be sent to reviewers for further evaluation.

Sincerely,

Sean T Prigge

Guest Editor

PLOS Pathogens

Xin-zhuan Su

Section Editor

PLOS Pathogens

Kasturi Haldar

Editor-in-Chief

PLOS Pathogens

orcid.org/0000-0001-5065-158X

Michael Malim

Editor-in-Chief

PLOS Pathogens

orcid.org/0000-0002-7699-2064

As you will see from the comments, all of the reviewers appreciate the work, the quality of the data and the contribution to the field. Reviewers 1, 2 and 3 suggest making relatively minor revisions, most of which are aimed at enhancing and clarifying different aspects of the results. Reviewer 4 made several suggestions and brought to your attention an article that was published today concerning PfPanK1 activity. In your response to the suggestions made by Reviewer 4, please pay specific attention to these points:

- Provide specific activity units in Figure 1c (as done in Fig 2).

- Determine the PanK activity of material (IP or lysate) from the Tg line TgPanK1-GFP/TgPanK2-mAIDHA after inducing knockdown of PanK2.

- Provide information about whether Tg14-3-3I can be detected with pan anti-14-3-3 in the TgPanK complex. Reviewer 1 also brought up this point.

Reviewer's Responses to Questions

**Part I - Summary**

Reviewer #1: The manuscript from Tjhin and Howieson et al., describes the role of pantothenate kinase (PanK) in both P. falciparum and T. gondii, two pathogenic apicomplexan parasites. While PfPanK1 has been previously characterized, this manuscript shows that PfPanK1 forms a complex with PfPanK2 and Pf14-3-3I and that PfPanK2 is also a functional kinase within the complex. Furthermore, this manuscript characterizes the homologues TgPanK1 and TgPanK2 using conditional knockdowns that show that the two proteins are essential for growth, form a complex and contribute to PanK activity.

I have some minor considerations which I list out below:

1. In Figure 2C, right panel, in which we observe the immunoprecipitation of the Pf14-3-3 protein. Here I think it is important to include the Western blots that show that immunoprecipitation of the GFP tagged PfPanK1 and PfPank2 for the experiment as well as the GFP control. Please provide all relevant panels of the immunoprecipitation experiment so that the evaluation of the experiment is complete.

2. The authors speculate that the TgPanK1 and TgPanK2 complex also harbors another yet unidentified protein due to the increase in molecular weight. Did the authors try the pan 14-3-3 antibody on these T. gondii extracts? If a corresponding 14-3-3 homolog does not exist in T. gondii, it would be good to mention this in the text. If one does exist, it would be good to discuss whether the 14-3-3 antibody detected anything within the TgPanK1/2 complex, even if it was a negative result.

3. PanOH-A, PanOH-B and CJ-A harbour mutations in PfPanK1 and has been previously characterized. Nevertheless, it would have been helpful in the text to list the mutations in PfPanK1 that are present within these strains.

4. It is clear that TgPanK1 and TgPanK2 are essential for proliferation. Would the authors be able to provide any additional information about the nature of the death using cell biology techniques?

5. Can the authors comment on why they used SaPanK for the compensate for the PanK loss in the conditional knockdown vs using PfPanK1 or PfPanK2?

Reviewer #2: The authors present a comprehensive analysis of a previously unreported heterodimeric PanK present in two apicomplexan parasites. A combination of biochemical, proteomic, and cell biological techniques deomstrates the activity of this new form of PanK complex and confirms that both PanK proteins are essential for PanK activity and parasite survival. Overall, the work is thorough and complete. It is well presented, and the data supports the authors’ conclusions.

Given the quality of the work it is a bit disappointing that the explanation of the broader significance of the findings is not more explicit. It is implied that the study is of wider importance because the novel, heterodimeric nature of the apicomplexan PanKs might provide insight into the purpose of “non-functional” and non-canonical PanKs found in other organisms. This idea is not developed further in the discussion with, for instance, a comparison of the structure of “non functional” PanKs in apicomplexans and other organisms and a discussion of whether heteromeric PanKs might be biologically important beyond the apicomplexans. The importance of the heteromeric PanKs in anti-parasitic drug development is only briefly mentioned, with no discussion on the impact this new PanK structure might have in the context of known inhibitors and their development as new antimalarials. A revision of the Introduction and Discussion to provide a deeper insight into these topics would provide the non-expert reader a better understanding of why this previously unreported form of PanK is of general significance.

Reviewer #3: The manuscript “A novel heteromeric pantothenate kinase complex in apicomplexan parasites” by Tjhin et al describes functional characterization of a unique heteromeric pantothenate kinase in P. falciparum and T. gondii which is formed by association of PanK1 and PanK2 proteins. By using a combination of fluorescent microscopy, immunoprecipitation, mass spectroscopy and SDS-PAGE or native PAGE the authors demonstrate that both PfPanK1 and PfPanK2, along with another protein, Pf14-3-3I are part of pantothenate kinase complex. This was further supported by studies involving episomal expression of PfPanK2 in pantothenate kinase mutant strains of P. falciparum. Similar observations regarding heteromeric complex of PanK1 and PanK2 were made in another parasite T. gondii by using an auxin inducible degron fusion and immunoprecipitation techniques. A combination of knockdown and heterologous complementation by S. aureus type II Pank established the essentiality of PanK1 and PanK2 for the proliferation of T. gondii parasites.

P. falciparum and T. gondii are important pathogens causing human disease and there is an urgent need to discover drugs with novel mechanism of action for controlling these diseases. Pantothenate kinase has been suggested to be a drug target in a number of bacterial pathogens. The results of this study suggest that pantothenate kinase is a potential drug target in these two parasites as well. The formation of heteromeric enzyme complex by association of PanK1 and PanK2 also explains why individually expressed PanKs in heterologous systems had not shown any catalytic activity in a number of earlier studies.

This is a scientifically sound study which has clearly established the essential nature and catalytic activity of pantothenate kinase in P. falciparum and T. gondii. The conclusions drawn have been duly supported with proper experimentation. All the experiments have been designed logically with proper controls whenever required. The publication of this study will open up new avenues in pantothenate kinase research and will help in studies involving functional characterization of hitherto uncharacterized pantothenate kinases in prokaryotes and eukaryotes.

Reviewer #4: The authors aimed at demonstration of the significance of the presence of two PanK in apicomplexan parasites. They demonstrated that PfPanK1 and 2 likely form a dimer of 240 kDa by BN-PAGE, and that the complex possesses kinase activity toward pantothenate. The authors reciprocally identified PfPanK1 and PfPanK2 by immunoprecipitation from GFP-PfPanK1 or GFP-PanK2-expressing transformant lines with comparable efficiency. Notably 14-3-3I (and also M17 leucyl aminopeptidase) were also detected at the significant level, although its significance was not further examined. They created GFP-PfPanK expressing lines also in PanK-inhibitor-resistant lines (PanOH-A, PanOH-B, and CJ-A), which carry mutations in PfPanK1, but not PfPanK2 (?), gene. The ranking of the strains based on the PanK activity detected in the lysate of both control and GFP-PanK1 expressing tranformant line in wildtype and three resistant strains are the same as the ranking when immunoprecipitated samples were used as enzyme source, which only circumstantially supported PanK1 and GFP-PanK2 interaction. However, this observation can be also explained by possible up- or down-regulation of PfPanK1/2 gene expression, post-translational modifications, and interaction with other accessary proteins (such as those detected by MS analysis of the immunoprecipitated samples).

**Part II – Major Issues: Key Experiments Required for Acceptance**

Reviewer #1: None.

Reviewer #2: (No Response)

Reviewer #3: (No Response)

Reviewer #4: This is carefully designed and well executed experiments in order to understand the conundrum that some organisms possess two PanK-like proteins, using two representative apicomplexan parasites, Pf and Tg, as examples. Discovery of the complex formation and essentiality of the both PanK isotypes for PanK activity and proliferation (though the essentiality in Pf was previously demosntrated) is a major contribution of the study to the fields. However, how the regulation of PanK activity by PanK2, e.g., via modulation of stability or protein complex conformation, remains totally undetermined. Furthermore, its relevance to pathogenesis of the two parasites are also not properly discussed. These two issues are more or less the critical points that need to be clarified. In addition, two additional critical missing points here are: they did not demonstrate stoichiometry of the two isotypes in the complex (e.g., the ratio of the components). Activity was simply shown as arbitrary (?) radio isotope counts, but not in specific activity (e.g., Fig. 1C), thus it was not evaluated that the complex is as active as detected in the whole cell lysate. In addition, PanK activity was not demonstrated using individual isoform of PfPanK1 and 2. The following recent paper demonstrated PanK1 activity using a single E. coli recombinant protein (not as a heterocomplex with PanK2) (Nurkanto et al., Front Cell Inf Microbiol, 2021; doi: 10.3389/fcimb.2021.639065). Thus, the biological significance of the heterodimer of PfPanK can be demonstrated using the two PfPanK recombinants.

Specific points:

Fig. 1b: Both PfPanK1-GFP and PfPanK2-GFP appear to be present as multiple truncated and untruncated forms (e.g., PfPanK1-GFP, ca 30, 38, 49, and 60kDa and 70-80 kDa untruncated form)(PfPanK2-GFP, ca 40, 60 kDa and -100kDa). It is important to know if such truncated forms also retains binding ability with PanK1 and 14-3-3I, and other possible accessory factors, and moreover if the truncated forms are also retain activity. The authors must address those points using recombinant proteins.

It is also important to note there is an additional complex (?) at -140kDa detected in the sample immunoprecipitated from PfPanK1-GFP only. After all, it is not clear at all which of the two isotypes is responsible for catalytic and regulatory? activities. Based on Tg KD experiments, it was shown that both isotypes are necessary for Tg proliferation. How about PanK activity? Specific KD of TgPanK1 or 2 reduces PanK activity in a time-dependent fashion?

Significance of interaction with 14-3-3I was intriguing; however, no further evidence for a potential involvement of 14-3-3I in influence on PanK activity, its synthesis and turnover, have not been provided. Neither was for other PanK1/2 associated proteins. Proteome data were not fully utilized and downstream research is rather superficial and sufficient for publication in PLoS Pathogens. They proposed in discussion section that Pf14-3-3I plays a regulatory role in the PfPanK complex. At least, conditional knockdown by GlnS-ribozyme-based conditional KO or destabilizing domain-based KD of Pf14-3-3I can be tried to see if PfPanK1/2 gene expression, protein localization, and function are affected. Furthermore, since 14-3-3I was not identified by immunoprecipitated from Tg, generalization of the possible interaction of PanK with 14-3-3I in apicomplexa should not be made. It was argued that the proteome of TgPanK was unsuccessful, but the whole content of the manuscript was intended to generalize the heteromeric formation of PanK for its activity, so it is indeed a pity if the proteomic analysis of TgPanK complex is not included in this study. This referee feels it is needed to include this for the completion of the present study if they generalize the observation for apicomplexa.

**Part III – Minor Issues: Editorial and Data Presentation Modifications**

Reviewer #1: Please see Part 1 summary

Reviewer #2: I addition to a revision of the Introduction and Discussion mentioned above, I would suggest some editing of the section describing the PfPanK immunoprecipitations. The experiments are excellent but the naming of the recombinant proteins and the multiple uses of GFP tags to pull down different complexes can make the details hard to follow. A minor reworking of this section and figures to emphasize which PanK is being expressed in each cell line (the type of tag is not as important), which protein is being immunoprecipitated, and which is being co-immunoprecipitated, would make the experiments easier to follow and the results easier to interpret. The inclusion of a diagram(s) in Figure 1/2 outlining the experimental approach would be helpful. While the description of the T. gondii immunoprecipitations was clearer, a diagram of these experiments would also be helpful.

Minor edits:

Line 41-43 should read “Interestingly, many eukaryotes (such as Arabidopsis thaliana 11,12, Mus musculus 13-16 and Homo sapiens 17-21) express multiple PanKs.”

Line 61: the comma after PfPank2 isn’t needed.

Line 341: “likely” can be deleted.

Figure 2a: the inclusion of M17 leucyl-aminopeptidase as the fourth most abundant protein in the text and in column two is a bit misleading. There are other proteins with more peptides recovered in the PanK2 immunoprecipitation and this should be made clearer.

Reviewer #3: (No Response)

Reviewer #4: The title seems to be a bit too general, and the observations should not be generalized to the apicomplexa as evidence was provided only on P. falciparum and T. gondii. It would be nice if the authors touch upon other apicomplexan parasites such as Eimeria spp., Theileria spp., and Cryptosporidium?

Figure 1C: Is it possible to prolong incubation to reach phosphorylation close to 100%?

Figure 2C: In control, the lysate of the line expressing GFP only did not show any PanK activity (pantothenate phosphorylating) up to 90 min. However, it is thought that in control, the endogenous PanK activity must be detected in the lysate (maybe low activity). Please explain.

“PfPanK1 and PfPanK2 are part of an active PanK enzyme complex in P. falciparum parasites”. However, one can also argue that PfPanK and 2 make a complex, but only PfPanK1 possesses an enzymological and physiological role. The evidence to claim the statement need to be provided (for instance, make a Pf mutant by removing only one of PfPanK1 and PfPanK2 genes)(if genes not being essential, though), to detect any decrease of PanK activity.

5. Fig 3B clearly showed that TgPanK1 and TgPanK2 play independent but cooperative roles for pantothenate phosphorylation. Provide any explanation on any expected specific roles of TgPanK1 and 2 in PanK activity in this parasite?

PLOS authors have the option to publish the peer review history of their article (what does this mean?). If published, this will include your full peer review and any attached files.

Reviewer #1: **Yes: **Wai-Hong Tham

Reviewer #2: No

Reviewer #3: **Yes: **Umender Sharma

Reviewer #4: No
---

## [Editor Report · Decision Letter 1]

13 Jul 2021

Dear Professor Saliba,

We are pleased to inform you that your manuscript 'A novel heteromeric pantothenate kinase complex in apicomplexan parasites' has been provisionally accepted for publication in PLOS Pathogens.

Best regards,

Sean T Prigge

Guest Editor

PLOS Pathogens

Xin-zhuan Su

Section Editor

PLOS Pathogens

Kasturi Haldar

Editor-in-Chief

PLOS Pathogens

orcid.org/0000-0001-5065-158X

Michael Malim

Editor-in-Chief

PLOS Pathogens

orcid.org/0000-0002-7699-2064
---

## [Editor Report · Acceptance letter]

23 Jul 2021

Dear Professor Saliba,

We are delighted to inform you that your manuscript, "A novel heteromeric pantothenate kinase complex in apicomplexan parasites," has been formally accepted for publication in PLOS Pathogens.

Best regards,

Kasturi Haldar

Editor-in-Chief

PLOS Pathogens

orcid.org/0000-0001-5065-158X

Michael Malim

Editor-in-Chief

PLOS Pathogens

orcid.org/0000-0002-7699-2064